# Cancer cachexia in *STK11/LKB1*-mutated non-small cell lung cancer is dependent on tumor-secreted GDF15

Jinhai Yu [1,2,3,12], Tong Guo[1,12], Arun Gupta [1], Ernesto M. Llano[1,3], Thomas Salisbury [4], Naureen Wajahat[1], Dianne Zhao [1], Sean Slater[1], Qing Deng[5], Esra A. Akbay [5], Beverly A. Rothermel [3], John M. Shelton [3], Bret M. Evers [5], Zhidan Wu[6], Iphigenia Tzameli[6], Evanthia Pashos[6], James Kim[3,4], John D. Minna [2,3,4,7], Puneeth Iyengar [8,9,10,13] ✉ & Rodney E. Infante [1,2,3,11,13] ✉

Cachexia is a wasting syndrome involving adipose, muscle, and body weight loss in cancer patients. Tumor loss-of-function mutations in *STK11/LKB1*, a regulator of AMP-activated protein kinase, induce cancer cachexia (CC) in preclinical models and are linked to weight loss in non-small cell lung cancer (NSCLC) patients. This study examines the role of the integrated stress response (ISR) cytokine growth differentiation factor 15 (GDF15) in regulating cachexia using patient-derived and engineered *STK11/LKB1*-mutant NSCLC lines. Tumor cell-derived serum GDF15 levels are elevated in mice bearing these tumors. Treatment with a GDF15-neutralizing antibody or silencing *GDF15* from tumor cells prevents adipose/muscle loss, strength decline, and weight reduction, identifying tumors cells as the GDF15 source. Restoring wild-type *STK11/LKB1* in NSCLC lines with endogenous STK11/LKB1 loss reverses the ISR and reduces GDF15 expression rescuing the cachexia phenotype. Collectively, these findings implicate tumor-derived GDF15 as a key mediator and therapeutic target in *STK11/LKB1*-mutant NSCLC-associated cachexia.

An understanding of the tumor and host genetic drivers of cancer cachexia (CC) remain elusive. This knowledge would be relevant from both a biomarker and therapeutic perspective. To overcome some of these knowledge barriers, we previously reported findings from a screen of human NSCLCs assessing their capacity to induce cachexia-associated fat and muscle wasting in immunodeficient mice[1]. We dichotomized a cohort of patient-derived NSCLC lines with CC-inducing capacity in vivo and a matched cohort of NSCLC lines without CC-inducing capacity. Interestingly, 80% of the CC-inducing NSCLCs possessed loss-of-function variants in *STK11/LKB1*, the upstream regulator of the cellular energy sensor AMP-activated protein kinase (AMPK). We subsequently showed that *STK11/LKB1* is a driver of cachexia development by silencing its expression in patient-derived and mouse tumor lines and demonstrating host wasting after

[1]Center for Human Nutrition, University of Texas Southwestern Medical Center, Dallas, TX, USA. [2]Harold C. Simmons Comprehensive Cancer Center, University of Texas Southwestern Medical Center, Dallas, TX, USA. [3]Department of Internal Medicine, University of Texas Southwestern Medical Center, Dallas, TX, USA. [4]Hamon Center for Therapeutic Oncology Research, University of Texas Southwestern Medical Center, Dallas, TX, USA. [5]Department of Pathology, University of Texas Southwestern Medical Center, Dallas, TX, USA. [6]Internal Medicine Research Unit, Pfizer Inc., Cambridge, MA, USA. [7]Department of Pharmacology, University of Texas Southwestern Medical Center, Dallas, TX, USA. [8]Department of Radiation Oncology, University of Texas Southwestern Medical Center, Dallas, TX, USA. [9]Department of Radiation Oncology, Memorial Sloan Kettering Cancer Center, New York, NY, USA. [10]Druckenmiller Center for Lung Cancer Research, Memorial Sloan Kettering Cancer Center, New York, NY, USA. [11]Department of Molecular Genetics, University of Texas Southwestern Medical Center, Dallas, TX, USA. [12]These authors contributed equally: Jinhai Yu, Tong Guo. [13]These authors jointly supervised this work: Puneeth Iyengar, Rodney E. Infante. ✉e-mail: iyengarp@mskcc.org; rodney.infante@utsouthwestern.edu

subcutaneous transplantation into immunocompromised and syngeneic mice, respectively[1]. Others have shown that *STK11/LKB1* silencing in genetically engineered *Kras* mutant mice induce CC in a majority of mice[2]. Finally, in a cohort of nearly 250 advanced NSCLC patients, ctDNA-based liquid biopsy assays established a significant correlation between a patient having a tumor with *STK11/LKB1* functional variant and the presence of CC-associated wasting at cancer diagnosis[1].

Having shown that tumor *STK11/LKB1* loss-of-function mutation is not only a predictive and prognostic biomarker but also functionally involved in CC induction/progression, we sought to identify the downstream tumor-intrinsic molecules transducing mutant *STK11/LKB1* CC signals. A candidate approach evaluating tumor-secreted factors with the potential to influence host wasting that had a dependence on tumor STK11/LKB1 functional loss led us to investigate GDF15 as such a molecule. GDF15 is an inflammatory cytokine of the TGFβ super family with many pathophysiologic roles, including in metabolism where it is thought to promote anorexia and subsequent weight loss by direct signaling of the hindbrain through the glial-cell-derived neurotrophic factor family receptor α-like (GFRAL)[3–10]. Originally shown to be released after activation of the integrated stress response from tissue injury or nutrient deprivation, GDF15 has now been found to regulate a number of fundamental processes[11–13]. Secreted GDF15 is first produced as a pro-form of protein within cells and only after processing by proteases is it converted into a mature form that can be found in systemic circulation acting centrally on GFRAL-positive neurons in the hindbrain[13–15]. Systemically, GDF15-induced anorexia was previously thought to be relevant when triggered by platinum-based and other cytotoxic chemotherapies offered to cancer patients[16,17]. Elevated circulating concentrations of GDF15 are assumed to be a consequence of host tissue response to patient exposure to these cytotoxic agents or as a result of disease progression[16]. Recent translational investigations also support an association between elevated GDF15 serum levels and the development of NSCLC-associated and other cancer-associated cachexia[18–20]. Independent of central effects, there are now other suggestions that GFD15 may signal peripherally by regulating contractility and energy expenditure in skeletal muscle[21,22]. In addition, antibody/small molecule inhibitors of GDF15/GFRAL signaling are being assessed in clinical trials to block cancer-associated cachexia[16,23,24]. Whether circulating GDF15 primarily originates from tumor or host cells, and the dependencies of tumor systems on GDF15 action remain unknown.

Here, we took advantage of our previous efforts to characterize the cachexia-inducing potential of a cohort of patient-derived NSCLC lines to delineate a tumor's GDF15 dependence in inducing cachexia and in identifying tumor intrinsic (human) versus extrinsic (murine host) contributions to systemic GDF15 concentrations. Leveraging our well-annotated cohort of human NSCLC patient-derived xenografts with and without *STK11/LKB1* variants, we establish an association between tumor cell *GDF15* mRNA expression, elevated circulating tumor-derived GDF15 levels, and loss-of-function *STK11/LKB1* variants. Parallel use of GDF15 neutralizing antibody and tumor cell *GDF15* silencing in multiple *STK11/LKB1*-mutated patient-derived and murine *Kras*/*Trp53*/*Stk11* NSCLC lines transplanted subcutaneously or orthotopically into murine models abrogated lung cancer cachexia development. Reconstitution of wild-type *STK11/LKB1* into a *STK11/LKB1*-mutated patient-derived NSCLC line decreased the tumor cell integrated stress response resulting in reduced GDF15 expression, processing, and release; suppressing the CC phenotype. Collectively, our data supports the role of tumor-secreted GDF15 in mediating mutant *STK11/LKB1* NSCLC cachexia development.

## Results

### Characterization of GDF15 tumor and circulating levels across a NSCLC cachexia cohort

We previously identified 17 human non-small lung cancer (NSCLC) lines that produced an unambiguous phenotype of cachexia (*n* = 10) or non-cachexia (*n* = 7) when injected subcutaneously into immunodeficient mice[1]. Of the 10 lines that induced host wasting, 8 lines had genetic variations in *STK11/LKB1* with moderate to high predicted functional impact. In a candidate approach to identify GDF15's relevance to STK11/LKB1-mediated cachexia in NSCLC, we assessed tumor *GDF15* mRNA expression and systemic GDF15 serum concentrations in the context of host wasting in the cachexia (*red*) and non-cachexia (*black*) lines from this cohort (Fig. 1A). *GDF15* gene expression was assessed by quantitative real-time PCR (RT-PCR) using human (tumor cell-derived) and mouse (host cell-derived) primers to distinguish tumor cell and host cell changes, respectively. Expression of human *GDF15* mRNA was significantly elevated in a majority of the CC-inducing lines compared to non-cachexia-inducing lines (Fig. 1B), with the exception of the H1395 and H1993 cachexia-inducing lines that had comparable levels to the non-cachexia cohort. Mouse *Gdf15* mRNA expression in tumors, representing host cell contribution to tumor expression, was statistically unchanged between the cachexia and non-cachexia groups (Fig. 1C).

To evaluate serum concentrations of GDF15 from the mice transplanted with the human cancer lines, we first established human (Supplementary Fig. S1A) and mouse (Supplementary Fig. S1B) specific GDF15 ELISAs that only recognized human GDF15 (tumor cell-intrinsic source) and murine GDF15 (host cell source), respectively. Serum concentrations of human GDF15 were significantly elevated in mice injected with the cancer cachexia-inducing *STK11/LKB1*-mutated NSCLC lines compared to the non-cachexia wild-type *STK11/LKB1* NSCLC lines and the PBS control cohort (Fig. 1D), whereas the serum mouse GDF15 concentration was statistically unchanged between the cachexia, non-cachexia, and PBS control groups (Fig. 1G). Importantly, there was a strong correlation between tumor human *GDF15* mRNA expression and serum human GDF15 concentrations (Fig. 1E). By contrast, there was no significant association of tumor mouse *Gdf15* mRNA expression with serum mouse GDF15 concentrations (Fig. 1H). Finally, there was no significant correlation between serum human GDF15 levels and tumor volumes from the in vivo studies (Fig. 1F). Overall, this data suggested that loss of functional STK11/LKB1-driven NSCLC cachexia is correlated with increases in *GDF15* mRNA expression and serum concentrations that are derived from the tumor cells and not the host.

### Human *STK11/LKB1*-mutant NSCLC dependence on high circulating GDF15 concentrations for cachexia induction

We next sought to identify if NSCLCs capable of inducing cachexia secondary to *STK11/LKB1* loss were dependent on elevated circulating GDF15 to induce wasting. More specifically to establish that tumor cell-secreted GDF15 initiates cachexia induction, the H1573 NSCLC cell line was transduced with lentiviral CRISPR/Cas9 constructs containing guide RNA targeting *GDF15* (H1573[ΔGDF15]) to silence expression. We used the *STK11/LKB1* mutant, CC-inducing patient line H1573 since it exhibited the highest tumor *GDF15* mRNA expression (see Fig. 1B) and the second highest serum concentration of tumor-derived GDF15 (see Fig. 1D) when xenotransplanted subcutaneously into immunocompromised mice. As a control, the parental H1573 line was also infected with guide RNA directed against *GFP* (H1573[ΔGFP]). Immunoblot analysis of the engineered H1573 cells determined that *GDF15* was only silenced in the H1573[ΔGDF15] tumor cells compared to the parental H1573 and the H1573[ΔGFP] tumor cells (Fig. 2A). These three lines and a PBS vehicle control were subsequently injected subcutaneously into NOD/SCID mice followed by measurement of tumor growth, food intake, body composition, body weight, and grip strength over time. As shown in Fig. 2B, C, tumors derived from the parental H1573, H1573[ΔGFP], and H1573[ΔGDF15] cell lines showed similar tumor growth kinetics and tumor weights at sacrifice, respectively. Immunoblot analysis verified that tumors derived from the H1573[ΔGDF15] line were durably suppressed from expressing GDF15 (Fig. 2A). Compared to mice harboring

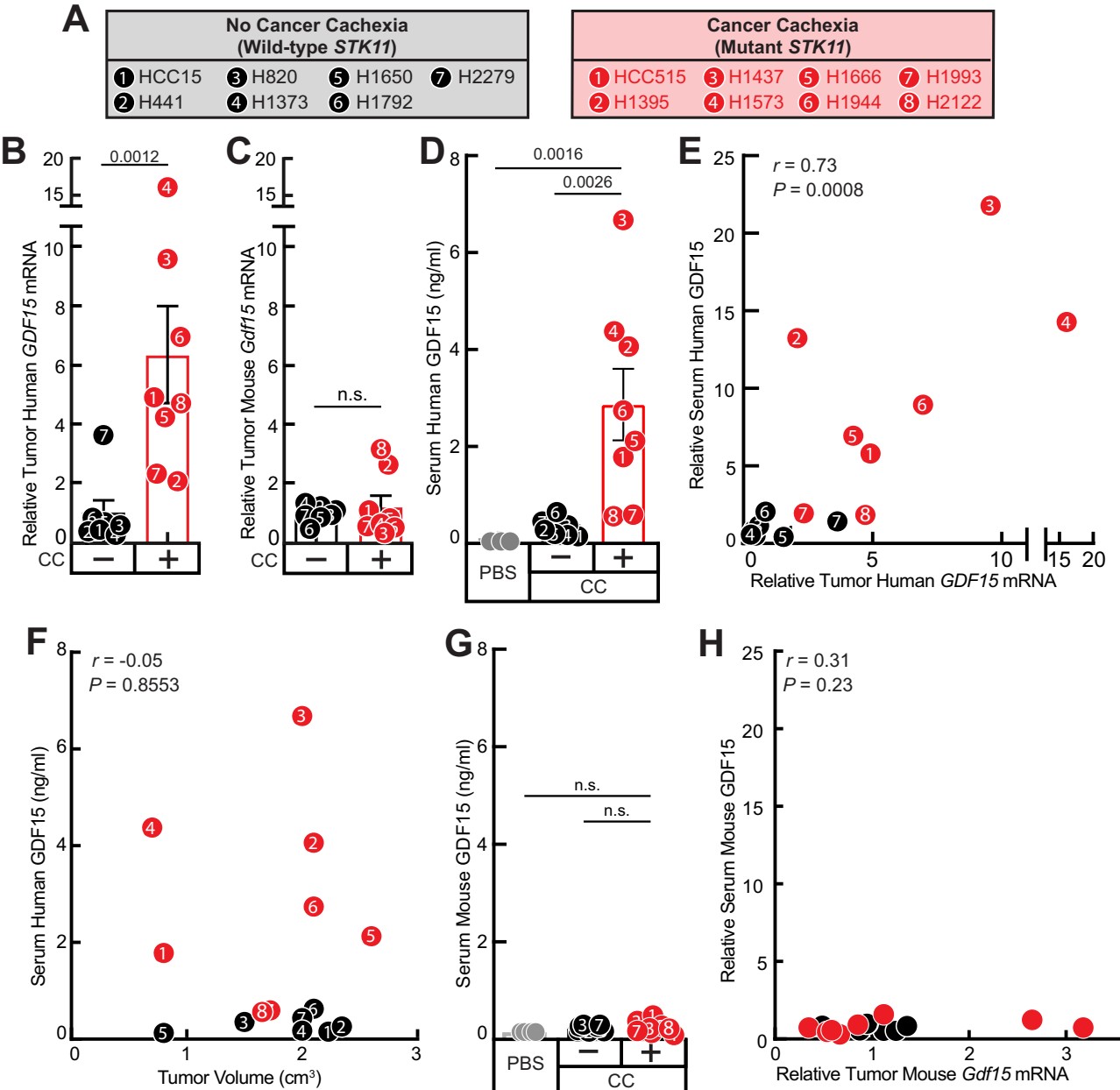

**Fig. 1 | Levels of mRNA and circulating GDF15 of human NSCLC cachexia. A** List of non-cachexia NSCLC lines with wild-type *STK11/LKB1* and cachexia NSCLC lines with mutant *STK11/LKB1* from which tumors and serum have been obtained from previous work when lines were xenotransplanted in vivo[1]. **B–H** Tumors were harvested, pooled and subjected to quantitative RT-PCR for human GDF15 (**B, E**) and mouse (**C, H**) *Gdf15* mRNA normalized to β-actin. Serum was subjected to the human (**D–F**) or mouse (**G, H**) GDF15 ELISA as described in Methods. Data are shown as mean ± SEM (**B–D, G**) or as a scatter plot (**E, F, H**) or relative to the average non-cachexia cohort (**B, C, E, H**) or the actual measurements (**D, F, G**). *P* was calculated based on unpaired 2-tailed *t* test (**B** and **C**) or 1-way ANOVA followed by Dunnett's multiple-comparison test (**D** and **G**), or from linear regression of *GDF15* mRNA levels with serum GDF15 level (**E, F, H**). n.s. not significant. 6 replicates for each point except *n* = 4 for H411, *n* = 3 for HCC15, *n* = 3 for H1944, *n* = 5 for H1573. Source data are provided as a Source Data file.

parental H1573 or H1573^ΔGFP tumors demonstrating serum GDF15 concentrations of ~4 ng/ml, serum from mice xenotransplanted with H1573^ΔGFP tumors had undetectable human (tumor-derived) GDF15 concentrations at levels comparable to findings in non-tumor bearing PBS-injected mice (Fig. 2D). ELISA measurements also revealed no significant changes in the serum mouse (host cell-derived) GDF15 concentrations in parental or engineered lines compared to the PBS control (Fig. 2E), suggesting a lack of host contribution to serum GDF15 levels.

Critically, compared to the mice injected with parental H1573 or H1573^ΔGFP cells, mice injected with the H1573^ΔGDF15 cell line had a rescue of their body weight (Fig. 2H), tumor-free body weight (Fig. 2I), lean

mass (Fig. 2K), tumor-free lean mass (Fig. 2L), and fat mass (Fig. 2J), similar to levels observed in the PBS-injected, non-tumor bearing mice. Mice bearing *STK11/LKB1*-mutated NSCLC tumors with intact GDF15 (parental 1573 and H1573^ΔGFP) demonstrated a significant decrease in grip strength, whereas mice bearing H1573^ΔGDF15 tumors had similar grip strength to PBS-injected non-tumor bearing mice (Fig. 2M). Measured another way, we provide in tabular form the ratio of grip strength to experiment day 0 body weight to generate a relative grip strength per mouse in each cohort (Supplementary Data 1).

Silencing of *GDF15* promoted minimal change in food intake in mice compared to the mice injected with parental H1573 or H1573^ΔGFP cells (Fig. 2G). At first approximation, this finding was counter-intuitive

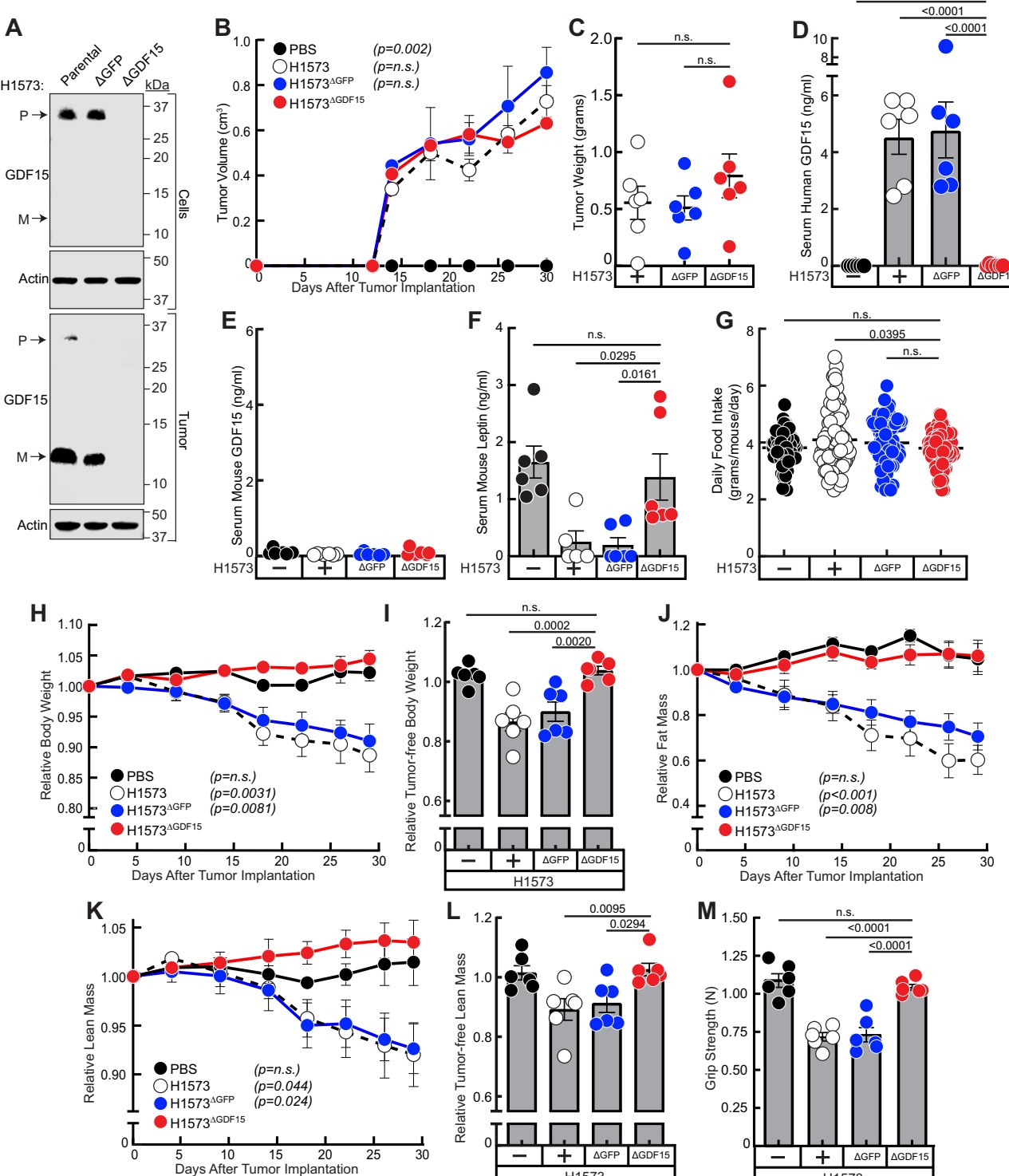

**Fig. 2 | *GDF15* silencing in *STK11/LKB1*-mutated NSCLC tumors subcutaneously xenotransplanted into mice that have high circulating GDF15 concentrations.**
**A**–**M** Chow-fed 13-week-old NOD/SCID male mice (*n* = 6 per group) were injected s.c. with 200 µl PBS in the absence or presence of 1 ×10⁷ cells from parental H1573, H1573^ΔGFP, or H1573^ΔGDF15 lines as described in Methods. Tumor cells before injection (cells) and tumors at sacrifice (tumor) were processed for immunoblot analysis with the indicated antibody, and the samples derive from the same experiment but different gels for GDF15 and Actin (**A**). Longitudinal or endpoint measurements of tumor volume (**B**), tumor weight (**C**), food intake (**G**), body weight (**H**), tumor-free body weight (**I**), fat mass (**J**), lean mass (**K**), tumor-free lean mass (**L**), and forelimb grip strength (**M**) were obtained as described in Methods. Serum at sacrifice was subjected to the human GDF15 (**D**), mouse GDF15 (**E**) or mouse leptin (**F**) ELISA as described in Methods. Data are shown as mean ± SEM of the actual measurements (**B**, **C**–**G**, **M**) or relative to their day 0 values (**H**–**L**). *P* was calculated using 1-way (**C**–**G**, **I**, **L**, **M**) or 2-way (**B**, **H**, **J**, **K**) ANOVA followed by Dunnett's multiple-comparison test for significant differences from the H1573^ΔGDF15 cohort. n.s. not significant; M = GDF15 mature protein; P = GDF15 proprotein. Source data are provided as a Source Data file.

to the function that GDF15 has been reported to have a role in regulating food intake. However, from a review of the literature, a lack of food intake change in GDF15-dependent cancer cachexia models was also observed by other investigators[23]. This group also failed to show increased food intake with suppression of GDF15-GFRAL signaling or significant suppression of food intake with elevated GDF15 serum concentrations. Earlier pre-clinical work by our group found that there are decreased levels of host systemic leptin to maintain food intake in the setting of tumor-secreted cachexia factors that typically decrease food intake and cause adipose wasting[25]. We next measured serum leptin concentrations to identify if this suggested compensation mechanism for food intake is observed in the GDF15-dependent H1573 cancer cachexia model. Figure 2F demonstrated that in control conditions of only PBS administration or with H1573$^{\Delta GDF15}$ cell administration, tumor bearing mice had baseline serum leptin concentrations corresponding to the presence of normal adipose levels. In the cohorts administered the 1573 parental or H1573$^{\Delta GFP}$ control cells, leptin concentrations had decreased significantly. The host compensation mechanism of decreasing serum leptin provides a possible explanation of why there are no food intake changes observed in this GDF15-dependent cancer cachexia model.

From a tissue perspective, there was protection against loss of both epididymal (Fig. 3A) and inguinal (Fig. 3B) white adipose tissue (eWAT and iWAT) mass in mice injected with H1573$^{\Delta GDF15}$ cells as compared to mice injected with parental H1573 or H1573$^{\Delta GFP}$ cells. Preservation of host adipose tissue with tumor cell *GDF15* silencing was further verified by H&E staining of epididymal white adipose tissue (Fig. 3C) and by measurement of average (Fig. 3D) or distribution of (Fig. 3E) adipocyte cross-sectional area showed similar adipocyte sizes to PBS-injected mice and significant larger adipocytes compared to those adipocytes taken from mice harboring parental H1573 or H1573$^{\Delta GFP}$ tumors. Both gastrocnemius (Fig. 3F) and quadriceps (Fig. 3G) weights were preserved similar to PBS-injected mice levels with tumor cell silencing of *GDF15*. Laminin staining of the gastrocnemius muscle revealed a significant increase of the total fiber cross-sectional area of the mice with the H1573$^{\Delta GDF15}$ tumors compared to mice harboring the parental or H1573$^{\Delta GFP}$ tumors (Fig. 3H, I). Finally, transcript levels of gastrocnemius and soleus *Fbxo32* and *Trim63* genes, both part of ubiquitin ligase complexes relevant to muscle atrophy[26], were negatively correlated with mouse body weight – i.e., their mRNA expression was elevated in mice bearing parental H1573 or H1573$^{\Delta GFP}$ tumor while being low in mice bearing H1573$^{\Delta GDF15}$ tumors or mice administered PBS - providing an explanation for the reduced muscle loss in this experimental arm of the study (Fig. 3J and Supplementary Fig. S2). Recently, a group highlighted the role of sarcolipin (*Sln*) in supporting GDF15 induced, food intake-independent body weight loss by regulating energy expenditure in skeletal muscle[22]. We, therefore, next evaluated the mRNA levels of muscle *Sln* in the H1573 cancer cachexia model. Gastrocnemius (Fig. 3J) and soleus (Supplementary Fig. S2) muscle mRNA levels of *Sln* expression were negatively correlated with mouse body weight. Muscle *Sln* was elevated in mice bearing *STK11/LKB1*-mutated tumors with intact GDF15, in cohorts of parental H1573 or H1573$^{\Delta GFP}$ tumor implanted mice. Muscle *Sln* expression remained low in mice bearing H1573$^{\Delta GDF15}$ tumors or mice administered PBS, providing an explanation for food intake-independent body weight loss in the GDF15 dependent H1573 cancer cachexia model. Overall, these studies demonstrated that *STK11/LKB1*-mutated H1573 NSCLC intrinsically rely on tumor cell-derived GDF15 for promotion of cachexia-associated adipose and muscle atrophy.

To identify any sexual dichotomy in cancer cachexia mechanisms and outcomes, we next conducted the same studies as depicted in Fig. 2 in the setting of female immunodeficient mice. Again, the H1573 NSCLC line was transplanted into mice unadulterated (parental control), with *GFP* silencing as a control (H1573$^{\Delta GFP}$), or with *GDF15* silencing as an experimental approach (H1573$^{\Delta GDF15}$). Though tumors grown

in the mice transplanted with the H1573$^{\Delta GDF15}$ line were on a slower growth trajectory than controls (Supplementary Fig. S3A), their final weights at sacrifice were comparable to controls and no longer statistically different (Supplementary Fig. S3B) in the female mice. Similarly to the findings in male mice, human serum GDF15 was suppressed to background levels compared to controls in female mice bearing H1573$^{\Delta GDF15}$ tumors (Supplementary Fig. S3C), with no significant level of mouse GDF15 measured in the serum of any cohort (Supplementary Fig. S3D). Phenotypically, GDF15-silenced H1573 tumor bearing mice demonstrated a preservation of body weight (Supplementary Fig. S3G), fat mass (Supplementary Fig. S3H), lean mass (Supplementary Fig. S3I), and grip strength (Supplementary Fig. S3J and Supplementary Data 1). From a tissue perspective, epididymal white adipose tissue (Supplementary Fig. S3K), inguinal white adipose tissue (Supplementary Fig. S3L), gastrocnemius (Supplementary Fig. S3M) and quadriceps (Supplementary Fig. S3N) tissue weights were also preserved in mice bearing H1573$^{\Delta GDF15}$ tumors vs control H1573 tumors. As conserved in male and female tumor cachexia murine models, we observed no decrease of food intake in the female mice bearing parental H1573 or H1573$^{\Delta GFP}$ tumors compared to PBS administrated mice and no increased food intake in mice bearing H1573$^{\Delta GDF15}$ tumors compared to the multiple cohorts of mice displaying cachexia (Supplementary Fig. S3F). The serum leptin concentrations decreased in mice with cachexia and remained stable in mice without cachexia as depicted in Supplementary Fig. S3E, again providing a possible explanation of why there was no net food intake changes observed in this GDF15-dependent cancer cachexia model leveraging female mice, paralleling findings from the 1573 experiment performed with male mice (see Fig. 2F). In summary, our cachexia findings were consistent across male and female mice, showing no obvious sexual dichotomy.

## Orthotopic modeling of *STK11/LKB1*-mutated NSCLC recapitulates the cachexia phenotype

We next conducted parallel studies with isogenic H1573 NSCLC lines by administering them orthotopically into the parenchyma of the left lung of immunocompromised mice as previously described[27–29]. Animals were administered either PBS, H1573$^{\Delta GFP}$ cells, or H1573$^{\Delta GDF15}$ cells in Matrigel and permitted to form foci of tumors in vivo as depicted in Fig. 4A. There was no difference in the ratio of tumor to lung area in H1573$^{\Delta GFP}$ or H1573$^{\Delta GDF15}$ tumor-bearing mice (Fig. 4B). As predicted, the mice bearing H1573$^{\Delta GFP}$ lung tumors demonstrated elevations in circulating human GDF15 compared to PBS- or H1573$^{\Delta GDF15}$-bearing mice (Fig. 4C). There was no murine GDF15 in circulation in any of the cohorts (Fig. 4D). As demonstrated in Fig. 4E, the H1573$^{\Delta GDF15}$-bearing mice did not demonstrate an increase in food intake when compared to the control cohorts. However, the H1573$^{\Delta GFP}$ tumor-bearing mice had a significant decrease in systemic leptin concentrations compared to the H1573$^{\Delta GDF15}$ tumor-bearing mice (Supplementary Fig. S8A), once again providing a possible explanation for the lack of change in food intake. All these findings translated into mice with H1573$^{\Delta GDF15}$ orthotopically placed lung tumors being protected against cachexia-associated body weight change (Fig. 4F), fat mass loss (Fig. 4G), and lean mass loss (Fig. 4H) compared to control cohorts. With respect to this last metric, silencing of *GDF15* in these lines suppressed quadriceps and gastrocnemius tissue weight loss (Fig. 4J) resulting in a preservation of grip strength (Fig. 4I and Supplementary Data 1) when compared to mice injected with control tumor cells or PBS alone. Staining of gastrocnemius tissue (Fig. 4L) and myocyte cross-sectional area evaluation (Fig. 4K) further demonstrated the maintenance of larger myocytes with *GDF15* silencing similar to PBS-injected mice and significantly increased compared to H1573$^{\Delta GFP}$-injected mice. With respect to more granular review of the adipose, the H1573$^{\Delta GDF15}$-bearing experimental mice were protected against epididymal and inguinal white adipose tissue loss in addition to protection from brown adipose tissue loss based on terminal tissue weight measurements (Fig. 4M).

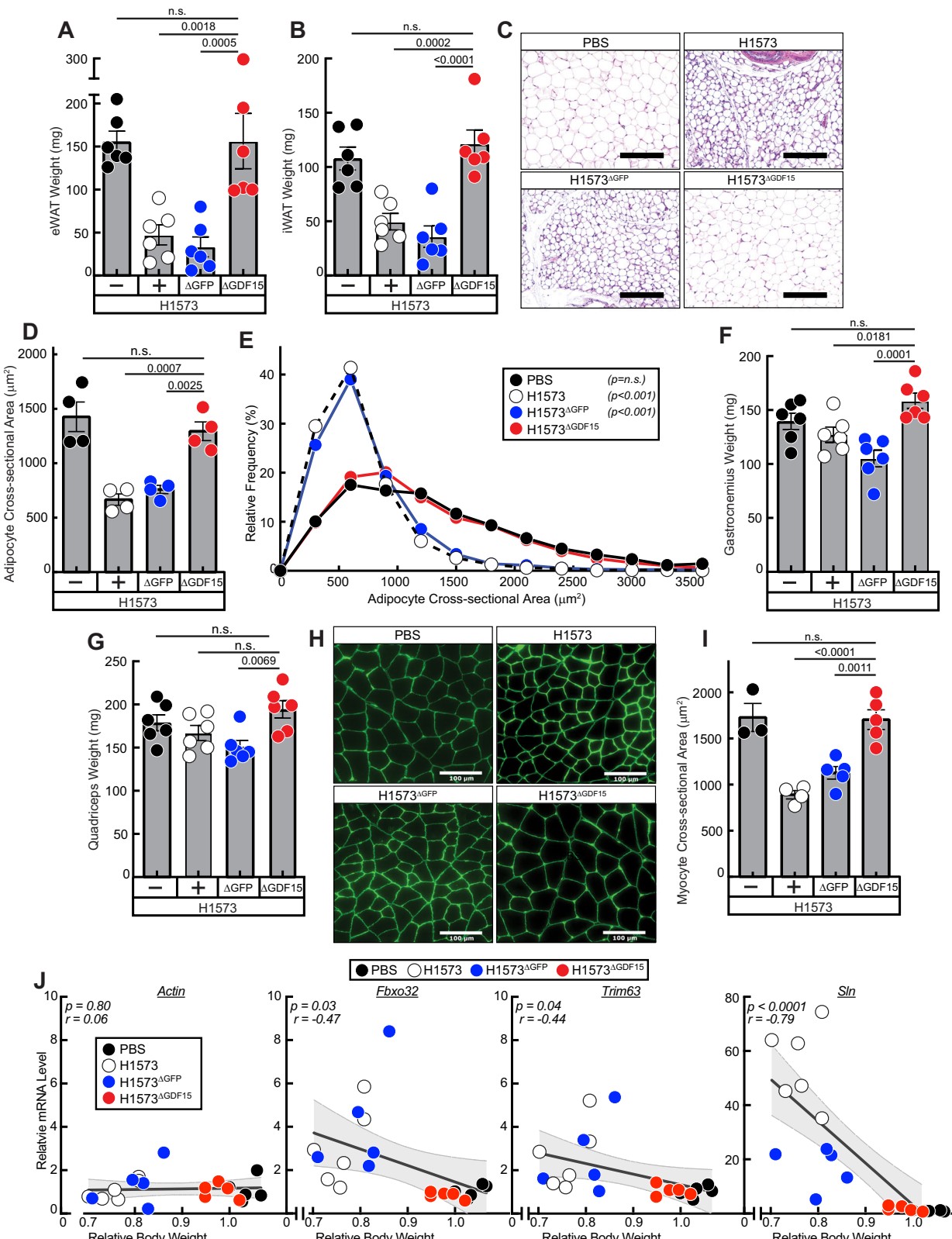

Sectioning and staining of epididymal white adipose tissue by H&E verified preservation of cell size, numbers, and architecture when *GDF15* was silenced (Fig. 4L), resulting in preservation of adipocyte cross-sectional area similar to PBS-injected mice and significantly increased compared H1573ΔGFP-injected mice (Fig. 4N). In total, we found that placement of cells to create tumor subcutaneously or orthotopically in the lungs solicited near identical cachexia development, and associated changes to muscle and adipose. These findings

are suggestive of the importance of tumor-secretory factors in regulating cachexia development from a para/endocrine approach irrespective of anatomic location.

### Antibody directed against GDF15 suppresses cancer cachexia-associated adipose and muscle wasting

The antibody against GDF15 (Pfizer murine analog to ponsegromab) used in our subsequent studies is reported to neutralize both human

**Fig. 3 | Analysis of adipose and muscle obtained from mice bearing isogenic H1573 NSCLC tumors. A–J** NOD/SCID mice were treated as described in Fig. 2 (*n* = 6 per group). Mice were sacrificed with epididymal white adipose tissue (eWAT) (**A**), inguinal white adipose tissue (iWAT) (**B**), gastrocnemius muscle (**F**), and quadriceps muscle (**G**) weighed. eWAT and gastrocnemius muscle were harvested and processed for H&E (**C**) or Laminin (**H**) staining, respectively (scale bar 100 μm). Subsequently, measurements of adipocyte area (**D, E**) (*n* = 4) and myocyte (**H, I**) (*n* = 3 for PBS, *n* = 4 for H1573, *n* = 5 for H1573^ΔGFP^ and H1573^ΔGDF15^) cross-sectional area were obtained. Gastrocnemius muscle was harvested and subjected to quantitative RT-PCR analysis of mouse mRNA levels of the indicated genes (*n* = 6 for non-tumor bearing mice group, *n* = 5 for tumor bearing mice group) (**J**). Data are shown as mean ± SEM of the actual measurements (**A, B, D, F, G,** and **I**), or as scattered plots with regression line and 95% confidence interval of relative values (**J**). *P* was calculated using 1-way (**A, B, D–G, I**) followed by Dunnett's multiple-comparison test for significant differences from the H1573^ΔGDF15^ cohort. *P* was calculated using Pearson correlation coefficients, two-tailed (**J**). n.s. not significant. Source data are provided as a Source Data file.

and murine orthologs[30]. As a control experiment, we showed that serial administration of the GDF15 antibody itself had no effect on wild-type mice with respect to changes in relative body weight (Supplementary Fig. S4A), fat mass (Supplementary Fig. S4B), and lean mass (Supplementary Fig. S4C), demonstrating nearly identical curves to mice receiving PBS or non-specific IgG. Subsequently, to continue testing the importance of tumor and/or host GDF15 to *STK11/LKB1*-associated NSCLC cachexia, we used a pharmacologic "prevention" approach by treating H1573 tumor-bearing mice with either vehicle (none), non-specific IgG monoclonal antibody, or the neutralizing antibody against GDF15 before any evidence of cachexia wasting and administered every 7 days over the span of the study (Fig. 5A). Another control in the experiment included a mouse cohort bearing H1573^ΔGDF15^ tumors, a model we verified above that cannot induce cachexia in vivo due to a lack of tumor-secreted GDF15. Administration of GDF15 neutralizing antibody or control IgG antibody had no effect on H1573 tumor growth (Fig. 5B) or weight (Fig. 5C) compared to parental H1573 or H1573^ΔGDF15^ tumor-bearing mice. Human ELISA analysis of serum demonstrated that the administration of GDF15 antibody significantly neutralized the serum human GDF15 concentrations from ~5 ng/ml in the vehicle- and IgG-treated H1573 mouse cohorts to ~0 ng/ml in mice (Fig. 5D), only matched in absence of circulating levels in mice bearing H1573^ΔGDF15^ tumors. Mice xenotransplanted with H1573 and administered GDF15 neutralizing antibody had a preservation of tumor-free body weight (Fig. 5E), fat mass (Fig. 5G), or tumor-free lean body mass (Fig. 5F) similar H1573^ΔGDF15^-bearing mice and significantly increased compared to vehicle- and IgG-treated H1573 tumor bearing mice. Again, non-specific IgG antibody treated mice were observed with lower serum leptin concentrations and GDF15 antibody administrated mice were observed with normal/baseline leptin concentrations (Supplementary Fig. S8B), providing a possible explanation for the lack of change in food intake. Overall, treatment with the GDF15 neutralizing antibody dramatically suppressed the development of cachexia-associated fat/lean mass and body weight loss normally induced by the *STK11/LKB1*-mutant H1573 NSCLC line.

We next conducted a cachexia "rescue" experiment by determining if GDF15 neutralizing antibody administration could stop and reverse cachexia associated wasting once muscle and fat losses had already been established. In this study, four cohorts of mice were employed – a control group administered only PBS, and three H1573 parental cohorts administered nothing, non-specific IgG antibody, and the GDF15 antibody. But, in this study, antibody or vehicle was only administered serially after 20% fat mass loss had been proven by longitudinal ECHO MRI (Fig. 5H). As anticipated, there was no difference in tumor growth kinetics (Fig. 5I) or weights (Fig. 5J). ELISA verified that the mice bearing H1573 tumors administered GDF15 antibody had a reduction in serum GDF15 to levels approaching non-tumor bearing control mice (Fig. 5K). Overall, GDF15 antibody administration even after the development of fat loss as a "rescue" strategy returned H1573-tumor bearing mice body weight loss (Fig. 5L), fat mass (Fig. 5M), epididymal white adipose weight loss (Fig. 5N), lean mass (Fig. 5O), and gastrocnemius tissue weight (Fig. 5P) back to levels similar to mice bearing no tumor and only administered PBS. Hence, it is apparent that the GDF15 neutralizing antibody can be used for prevention and/or rescue of cachexia induced by NSCLCs with *STK11/LKB1*

loss of function alterations promoting wasting through tumor GDF15 induction.

## Human *STK11/LKB1*-mutant NSCLC dependence on intermediate and low circulating GDF15 concentrations for cachexia induction

Having established that mice xenotransplanted with human NSCLC lines demonstrating high circulating human GDF15 concentrations (~4 ng/ml or more) are likely dependent on tumor-derived GDF15 for cachexia induction, we next sought to determine if intermediate serum concentrations of human GDF15 (~2 ng/ml) were sufficient to support cachexia-associated adipose and muscle wasting induced by NSCLCs possessing *STK11/LKB1* loss of function variants. The parental human 1944 NSCLC line had average human *GDF15* expression (see Fig. 1B) and intermediate serum concentrations (~2 ng/ml; see Fig. 1D) compared to the other *STK11/LKB1* mutant CC inducing lines. We therefore separately silenced *GDF15* (H1944^ΔGDF15^) or *GFP* (H1944^ΔGFP^) in the parental H1944 cells using lentiviral CRISPR/Cas9 techniques. Immunoblot analysis verified that that the monoclonal H1944^ΔGDF15^ line had silenced GDF15 protein expression compared to parental H1944 and H1944^ΔGFP^ control cell lines (Supplementary Fig. S5A). These parental and engineered cell lines were subsequently injected subcutaneously into NOD/SCID immunodeficient mice and tumors derived from the H1944^ΔGDF15^ cells showed persistent elimination of GDF15 protein expression by immunoblot analysis (Supplementary Fig. S5A). There was no significant change in tumor growth when comparing the parental H1944, H1944^ΔGFP^, and H1944^ΔGDF15^ tumors (Supplementary Fig. S5B). Serum from mice harboring H1944^ΔGDF15^ tumors had a significant decrease in human (tumor cell-derived) GDF15 concentration compared to the serum of mice injected with parental H1944 or H1944^ΔGFP^ cells (Supplementary Fig. S5C). However, serum mouse (host cell-derived) GDF15 concentrations remained low in all three cohorts (Supplementary Fig. S5D). In this model, mice harboring H1944^ΔGDF15^ tumors had a significant increase in their daily food intake (Supplementary Fig. S5E) with a preservation of tumor-free body weight (Supplementary Fig. S5F), tumor-free lean mass (Supplementary Fig. S5G), and fat mass (Supplementary Fig. S5H). These studies confirmed that intermediate concentrations of circulating GDF15 derived from an *STK11/LKB1*-mutated NSCLC line can also induce the cachexia phenotype, with the cachexia rescued by functional genetic targeting of tumor cell *GDF15* expression.

Using GDF15 neutralizing antibody and/or tumor cell *GDF15* silencing studies, we were able to clearly show that cancer cachexia models driven by *STK11/LKB1* loss of function tumor variants with high and intermediate concentrations of circulating GDF15 were dependent on this tumor-derived factor to promote cachexia-associated fat and muscle atrophy. We used the next set of studies to establish a lower GDF15 concentration boundary of dependence by *STK11/LKB1*-mutated NSCLC lines for promoting cachexia wasting. Although immunodeficient mice harboring *STK11/LKB1*-mutated H2122 tumors induce cachexia when injected subcutaneously into NOD/SCID mice, their human (tumor-derived) *GDF15* mRNA expression (see Fig. 1B) and GDF15 serum concentrations (see Fig. 1D) were at levels similar to mice bearing NSCLCs with wild-type *STK11/LKB1* that could not induce cachexia. Next, we silenced *GDF15* (H2122^ΔGDF15^) or *GFP* (H2122^ΔGFP^) in

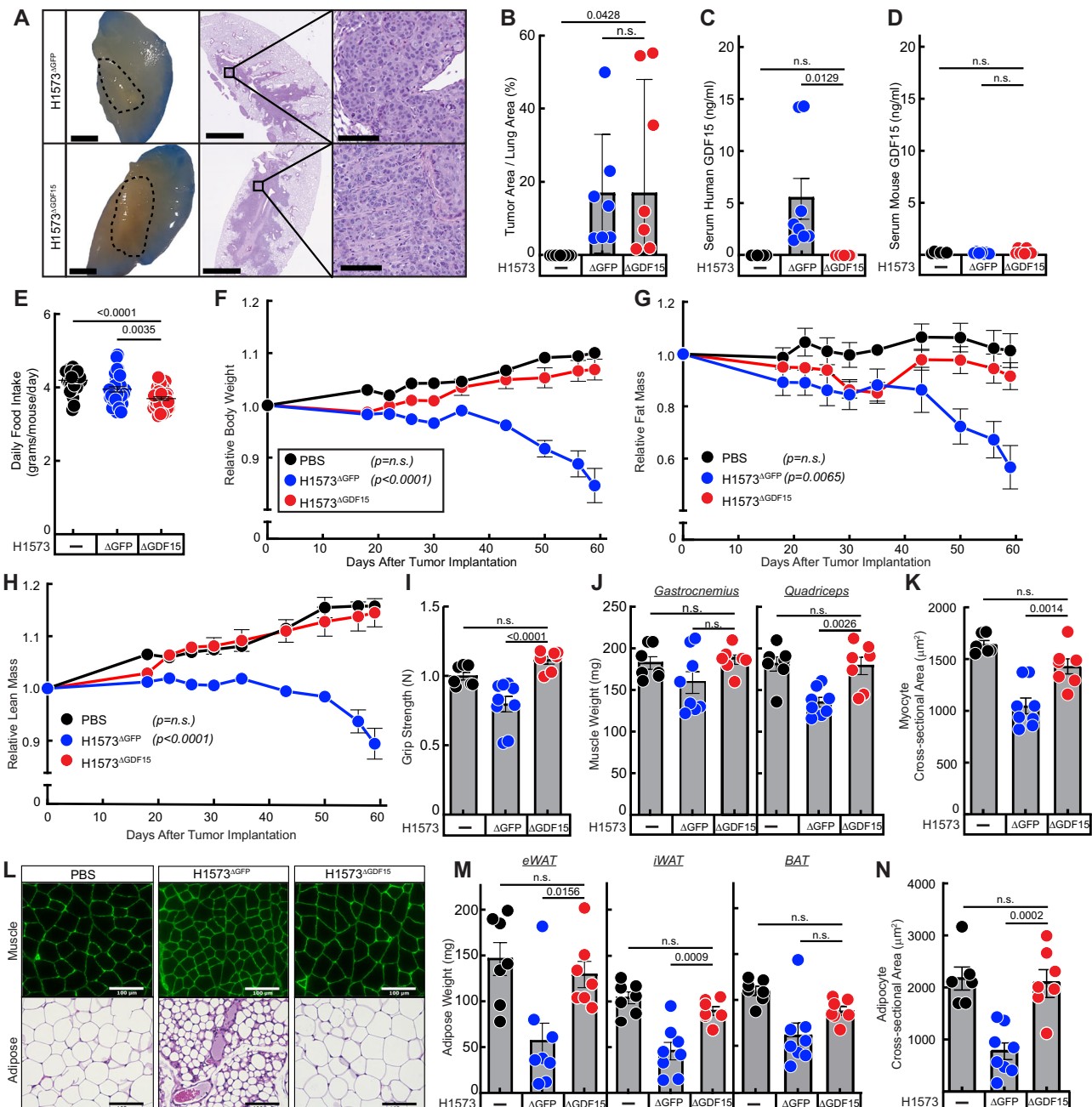

**Fig. 4 | GDF15 silencing in *STK11/LKB1*-mutated NSCLC tumors orthotopically xenotransplanted into mice that have high circulating GDF15 concentrations.** **A**–**N** Chow-fed 11–12-week-old NOD/SCID male mice (*n* = 9 per group) were injected to left lung with 30 μl of 25% growth factor reduced Matrigel in dye-free RPMI medium in the absence or presence of 1 ×10⁶ cells from H1573^ΔGFP, or H1573^ΔGDF15 lines as described in Methods. Longitudinal or endpoint measurements of tumor area (**A**, **B**), food intake (**E**), body weight (**F**), fat mass (**G**), lean mass (**H**), or forelimb grip strength (**I**) were obtained as described in Methods. Serum at sacrifice was subjected to the human (**C**) or mouse (**D**) GDF15 ELISA as described in Methods.

Indicated muscles (**J**) or fat pads (**M**) were weighed. eWAT and gastrocnemius muscle were harvested and processed for H&E or Laminin staining, respectively (**L**) (scale bar 100 μm). Subsequently, measurements of myocyte (**K**) and adipocyte (**N**) cross-sectional area were obtained (*n* = 6 for non-tumor bearing mice group, *n* = 8 for H1573^ΔGFP, *n* = 7 for H1573^ΔGDF15). Data are shown as mean ± SEM of the actual measurements (**B**, **C**–**E**, **I**–**K**, **M**, **N**) or relative to their day 0 values (**F**–**H**). P was calculated using 1-way (**B**–**E**, **I**–**K**, **M**, **N**) or 2-way (**F**–**H**) ANOVA followed by Dunnett's multiple-comparison test for significant differences from the H1573^ΔGDF15 cohort. n.s. not significant. Source data are provided as a Source Data file.

the parental H2122 cells using lentiviral CRISPR/Cas9 techniques. Immunoblot analysis revealed that the H2122^ΔGDF15 line had suppression of GDF15 protein expression compared to parental H2122 and H2122^ΔGFP cell lines (Fig. 6A). These cell lines were subsequently injected subcutaneously into NOD/SCID immunodeficient mice followed by measurement of the wasting phenotype. There was no significant difference in tumor growth among the three tumor cohorts (Fig. 6B) despite the persistent suppression of GDF15 protein expression from

the H2122^ΔGDF15 tumors (Fig. 6A). Tumor weight measurements conducted at study end showed H2122^ΔGDF15 tumor weights were similar to tumor weights compared to H2122 parental tumors, but significantly decreased compared to H2122^ΔGFP tumors (Fig. 6C). Mice harboring the parental H2122 or the H2122^ΔGFP lines had average serum human (tumor cell-derived) GDF15 concentrations less than 0.5 ng/ml (Fig. 6D). Although this was significantly increased compared to PBS-treated mice, this concentration of GDF15 was an order of magnitude lower

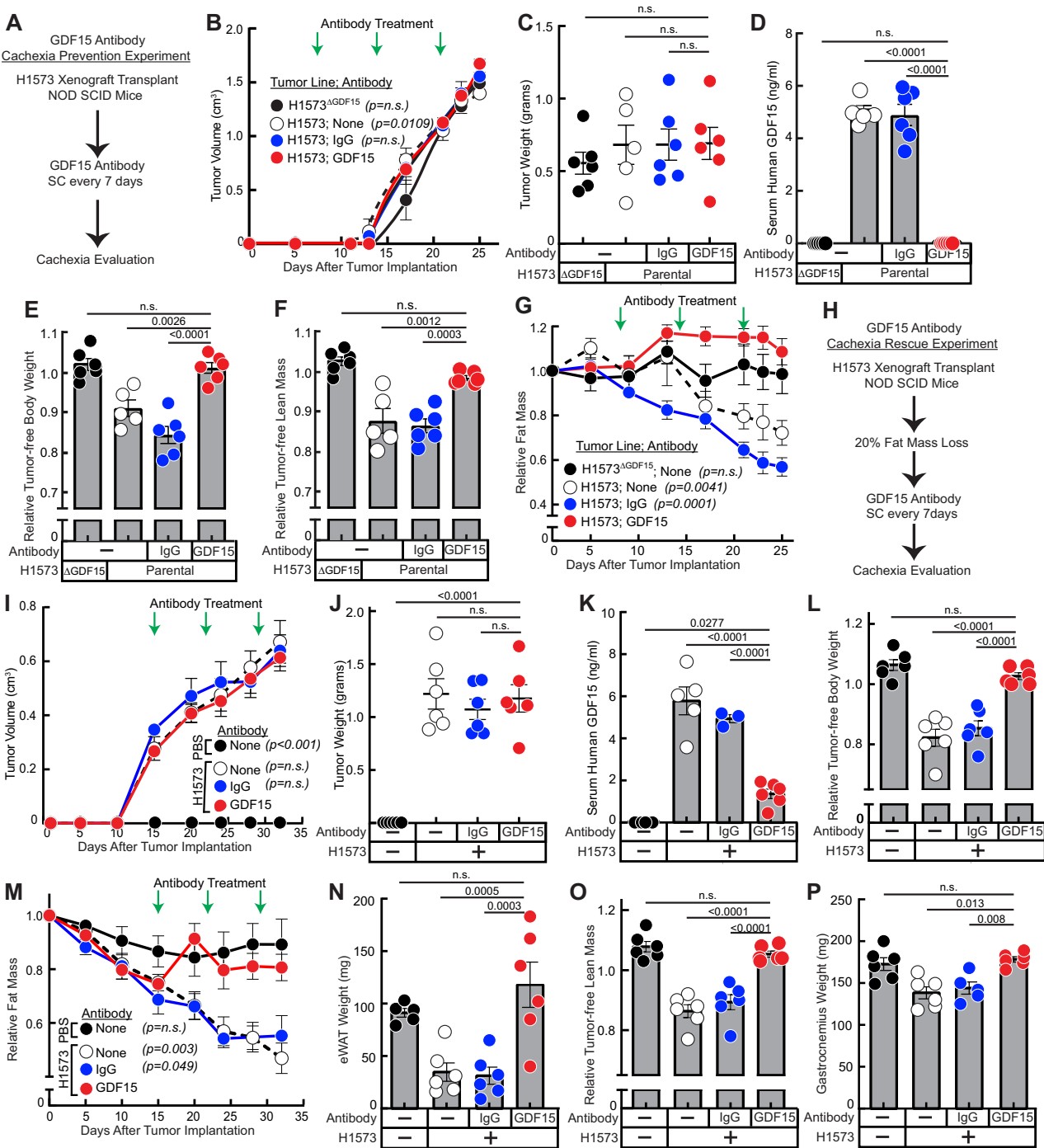

**Fig. 5 | GDF15 antibody neutralization in mice bearing *STK11/LKB1*-mutated NSCLC tumors with high circulating GDF15 concentrations. A–P** Chow-fed 12-13-week-old NOD/SCID male mice (*n* = 6) were injected s.c. with 200 μl PBS containing 1 ×10⁷ cells from parental H1573 human NSCLC line on day 0. **A–G** Starting on day 7 or day 15 (**H–P**), mice were injected with s.c. 150 μl PBS in the absence or presence of 10 mg/kg IgG or GDF15 antibody weekly. Longitudinal or endpoint measurements of tumor volume (**B, I**), tumor weight (**C, J**), tumor-free body weight (**E, L**), fat mass (**G, M**), tumor-free lean mass (**F, O**), eWAT weight (**N**), and Gastrocnemius weight (**P**) were obtained as described in Methods. Serum at sacrifice was processed for human GDF15 ELISA (**D, K**) as described in Methods. Data are shown as mean ± SEM of the actual measurements (**B–D, I–K, N, P**) or relative to their day 0 values (**E–G, L–M, O**). P was calculated using 1-way (**C–F, J–L, N–P**) or 2-way (**B, G, I** and **M**) ANOVA followed by Dunnett's multiple-comparison test for significant differences from the GDF15 antibody cohort. n.s. not significant. Source data are provided as a Source Data file.

than observed in the H1573 (see Fig. 2D and Supplementary Fig. S2C) mouse studies. As expected, mice harboring H2122ΔGDF15 tumors had a significant decrease in human (tumor cell-derived) GDF15 concentrations (Fig. 6D). Mouse GDF15 ELISAs showed no significant serum expression of GDF15 in the models tested (Fig. 6E). With respect to resulting phenotypes, we did observe a significant increase in food intake in the mice harboring H2122ΔGDF15 tumors compared to those mice bearing parental tumors (Fig. 6F). There was also a significant preservation of tumor-free body weight (Fig. 6G), tumor-free lean mass (Fig. 6H), and fat mass (Fig. 6K) in the experimental cohorts compared to the parental H2122- and H2122ΔGFP- bearing mice. More specifically, staining of muscle tissue (Fig. 6I) demonstrated larger myocyte cross-

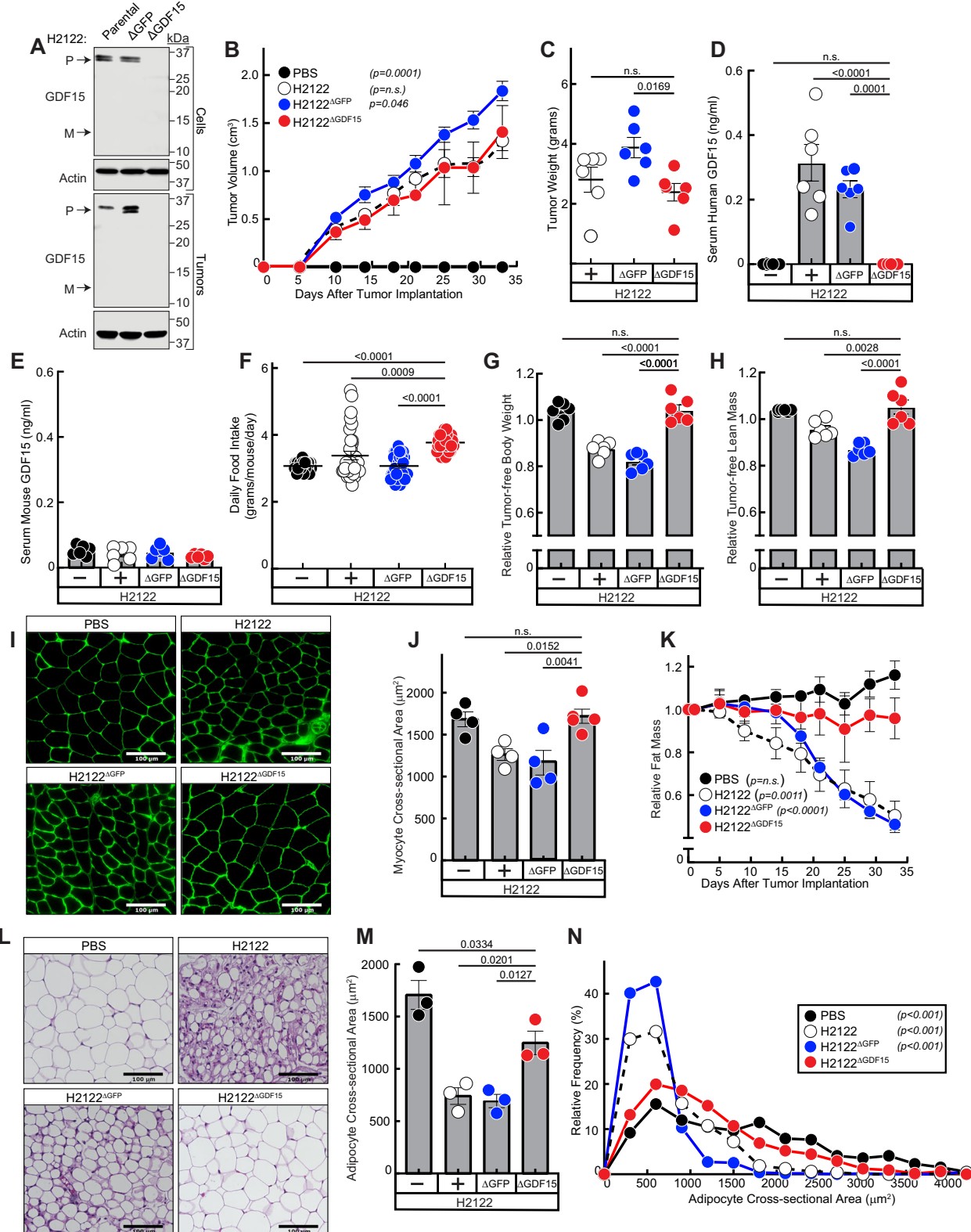

sectional area (Fig. 6J) in mice bearing *GDF15*-silenced tumors than in control tumor bearing cohorts. Similarly, H&E staining of adipose tissue (Fig. 6L) showed larger adipocyte cross-sectional area (Fig. 6M) and a distribution of areas shifted towards larger sizes (Fig. 6N) in mice bearing *GDF15*-silenced tumors. Overall, tumor-secreted GDF15 also contributes to the cachexia phenotype in mice bearing *STK11/LKB1*-mutated NSCLC lines demonstrating low systemic concentrations of the protein.

## Dependence of *STK11/LKB1*-mutated NSCLC cachexia on GDF15 is verified in syngeneic models

Previously, all experiments interrogating cachexia biology in the context of *STK11/LKB1* tumor loss and GDF15 relevance had prioritized using human NSCLCs derived from patients in immunodeficient murine models. To determine if similar findings could be recapitulated with an intact immune system, we obtained murine isogenic tumors cell lines derived from well-established KP and KPL genetically-

**Fig. 6 | *GDF15* silencing in *STK11/LKB1*-mutated NSCLC tumors xeno-transplanted into mice that have low circulating GDF15 concentrations.**
**A**–**N** Chow-fed 13-week-old NOD/SCID male mice (*n* = 6 per group) were injected s.c. with 200 μl PBS in the absence or presence of $1 \times 10^7$ cells from parental H2122, H2122$^{\Delta GFP}$, or H2122$^{\Delta GDF15}$ lines as described in Methods. Tumor cells before injection (cells) and tumors at sacrifice (tumors) were processed for immunoblot analysis with the indicated antibody, and the samples derive from the same experiment but different gels for GDF15 and Actin (**A**). Longitudinal or endpoint measurements of tumor volume (**B**), tumor weight (**C**), food intake (**F**), tumor-free body weight (**G**), tumor-free lean mass (**H**), or fat mass (**K**) were obtained as described in Methods. At sacrifice, serum was subjected to the human (**D**) or mouse (**E**) GDF15 ELISA and gastrocnemius muscle and epididymal white adipose tissue were harvested and processed for Laminin (**I**) or H&E (**L**) staining, respectively (scale bar 100 μm). Subsequently, measurements of myocyte (**J**) (*n* = 4 for non-tumor bearing mice group, *n* = 4 parental for H2122 and H2122$^{\Delta GFP}$, *n* = 5 for H2122$^{\Delta GDF15}$) and adipocyte (**M**, **N**) cross-sectional area from 3 mice per cohort were obtained as described in Methods. Data are shown as mean ± SEM of the actual measurements (**B**–**F**, **J**, **M**, **N**) or relative to their day 0 values (**G**, **H**, **K**). *P* was calculated using 1-way (**C**–**H**, **J**, **M**, **N**) or 2-way (**B**, **K**) ANOVA followed by Dunnett's multiple-comparison test for significant differences from the H2122$^{\Delta GDF15}$ cohort. n.s. not significant; M = GDF15 mature protein; P = GDF15 proprotein. Source data are provided as a Source Data file.

engineered mouse models (GEMMs)[31,32]. The KP model has mutations in *Trp53* and *Kras* leading to functional inactivation of both genes whereas the KPL model has an additional inactivating mutation in *Stk11/Lkb1*. Use of tumor cells derived from these murine lines in allotransplant experiments permitted us the ability to titrate the timing of tumor generation and kinetics of tumor growth in the context of a fully functional immune system. As expected, the KPL cell line expressed no STK11/LKB1 and had limited phosphorylation of AMPK (Supplementary Fig. S6A) compared to the KP line determined by immunoblot analysis. KPL tumors had a trend of increased growth kinetics (Supplementary Fig. S6B) and significantly increased tumor weights (Supplementary Fig. S6C) compared to KP tumors, a result observed by others and our previous work when silencing STK11/LKB1 in tumors transplanted into syngeneic models[1,31]. Food intake was reduced in the mice bearing KPL tumors compared to the control cohorts (Supplementary Fig. S6D). In Supplementary Fig. S6E, KPL-bearing mice demonstrated elevations in serum GDF15 concentrations above baseline/PBS-injected control animals. Although there was ~2-fold increase in serum mouse GDF15 levels in KPL-bearing mice compared to KP-bearing mice, these serum concentrations were lower than all of the patient-derived *STK11/LKB1*-mutated NSCLC lines previously tested (see Fig. 1D). Unlike tumors formed from the KP line, however, those formed from transplantation of the KPL line were able to induce host cachexia-associated tumor-free body weight loss (Supplementary Fig. S6F), tumor-free lean mass loss (Supplementary Fig. S6G), and fat loss (Supplementary Fig. S6H) in the mice into which they were transplanted.

To determine if GDF15 is relevant to cachexia induced in KPL-bearing animals, we suppressed GDF15 action with the administration of a GDF15 neutralizing antibody to specific KPL-bearing mouse cohorts. PBS and non-specific IgG were administered to the control KPL cohorts. These studies verified that KPL-induced wasting metrics – anorexia (Fig. 7C), tumor-free body weight loss (Fig. 7D), fat loss (Fig. 7E), and lean mass loss (Fig. 7H) were dependent on GDF15 despite similar tumor growth (Fig. 7A) and terminal tumor weights (Fig. 7B). Finally, we verified that the GDF15 antibody was able to suppress KPL-mediated, cachexia-associated decreased adipocyte (Fig. 7F, G) and muscle (Fig. 7I, J) cross-sectional areas. Overall, this syngeneic KP/KPL model further validated GDF15's role in *STK11/LKB1*-mutated cachexia induction even at low serum concentrations.

### Reconstitution of wild-type *STK11/LKB1* verifies its role in regulating cachexia through regulation of GDF15 expression and release

To assess if we could abrogate cachexia induction in *STK11/LKB1*-mutant NSCLC lines with reconstitution of wild-type STK11/LKB1 protein, we identified the STK11/LKB1-mutated H1437 line that does not make endogenous STK11/LKB1 at baseline due to a deletion of exons 2 and 3[1]. Tumor-bearing parental H1437 mice had the second highest tumor mRNA expression and the highest serum concentrations of human tumor-cell-derived GDF15 when compared to the other *STK11/LKB1*-mutant lines and the non-cachexia lines as well (see Fig. 1B, D). Again, we established the importance of GDF15 to this *STK11/LKB1*-

mutated NSCLC in vivo by silencing the expression of the cytokine with CRISPR/Cas9 techniques (Supplementary Fig. S7A), leading to the elimination of the protein in its precursor or mature form in cells or medium. Injection of mice with H1437$^{\Delta GDF15}$ cells had no effect on tumor volume (Supplementary Fig. S7B), tumor weight (Supplementary Fig. S7C), or increase in food intake (Supplementary Fig. S7F) even though there was suppression of human serum GDF15 levels (Supplementary Fig. S7D) with no effect on murine serum GDF15 levels (Supplementary Fig. S7E). H1437$^{\Delta GDF15}$ tumor bearing mice were protected against body weight loss (Supplementary Fig. S7G), lean mass loss (Supplementary Fig. S7H), and fat mass loss (Supplementary Fig. S7I). *GDF15* silencing also prevented reductions in myocyte (Supplementary Fig. S7K) and adipocyte cross-sectional areas (Supplementary Fig. S7L) compared to controls, measured from relevant muscle and adipose tissue staining, respectively (Supplementary Fig. S7J). The H1437$^{\Delta GFP}$ tumor-bearing mice had a significant decrease in systemic leptin concentrations compared to the H1437$^{\Delta GDF15}$ tumor-bearing mice (Supplementary Fig. S8C), once again providing a possible explanation for the lack of change in food intake in this model.

We next overexpressed the wild-type gene using a lentivirus system and made stable H1437$^{STK11}$ tumor lines. We also created a monoclonal control line that received only empty vector (H1437$^{CTL}$). As observed from immunoblotting analysis in Fig. 8A, we were successfully able to create two monoclonal lines overexpressing wild-type STK11/LKB1 (H1437$^{STK11}$), associated with the expected increase in phosphorylation of AMPK compared to the parental H1437 and the H1437$^{CTL}$ tumor lines. This finding was verified in the stable H1437$^{STK11}$ lines created in addition to the tumors derived from these lines in vivo. We next tested these cells in conditions of nutrient rich (high glucose) and nutrient deprived (low glucose) in vitro medium conditions, especially knowing that GDF15 mediates nutrient-limited states through the activation of integrated stress response (ISR) proteins CHOP and ATF4[13]. The endoplasmic reticulum ISR signaling pathway includes multiple molecules, including PERK activation and its phosphorylation of eIF2α, and subsequent nuclear ATF4 translocation supporting its transcriptional activity[13]. In the parental 1437 and H1437$^{CTL}$ NSCLC lines devoid of wild-type STK11/LKB1 expression, several gene products, including GDF15 and the ISR proteins CHOP and ATF4, were increasingly induced with decreasing glucose/nutrients (Fig. 8B). However, when wild-type STK11/LKB1 was reconstituted into the parental 1437 NSCLC line, decreasing medium glucose did not elicit an exaggerated ISR as observed by decreased ATF4 through immunoblot analysis and was associated with decreased cellular and secreted GDF15 (pro and mature forms). CHOP expression was relatively unchanged when comparing lines with wild-type and mutant STK11/LKB1 protein. Tumors derived from transplantation of the 1437 line reconstituted with wild-type STK11/LKB1 maintained expression of the kinase while still suppressing expression of all forms of GDF15. The tumor ISR signaling pathway was reduced as shown by a reduction in PERK levels and phosphorylation of the translation initiation factor eIF2 as expected (Fig. 8C). This data provided evidence of a clear mechanistic interaction between the presence or absence of STK11/LKB1 protein and the induction of the ISR with subsequent expression

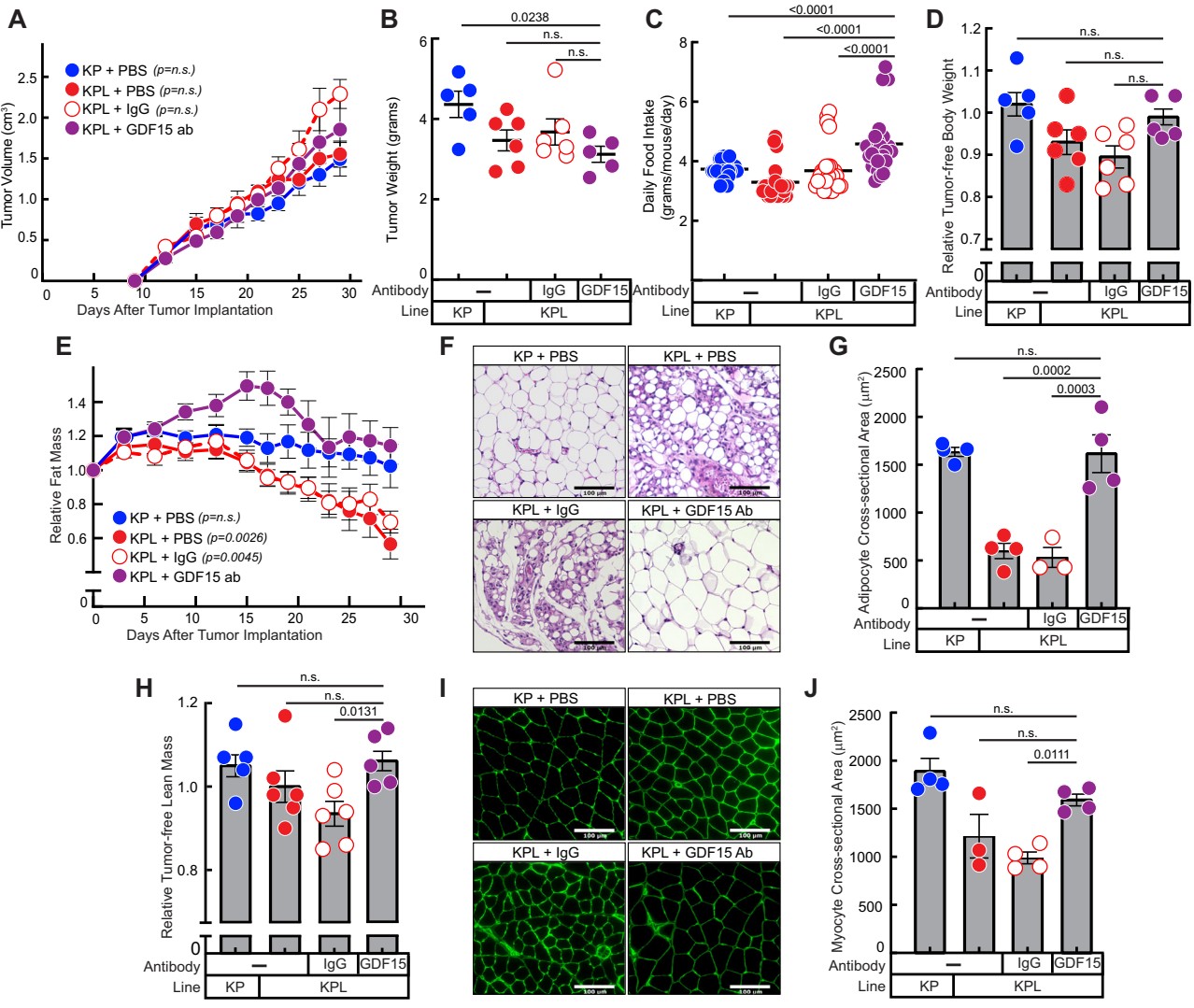

**Fig. 7 | GDF15 antibody neutralization in syngeneic mice bearing KP and KPL tumors. A–J** Chow-fed 12-week-old C57BL6J male mice (*n* = 6) were injected s.c. with 200 µl PBS containing in the absence or presence of 1 ×10⁶ cells from the mouse NSCLC KP or KPL lines on day 0 followed by weekly injections of 150 µl PBS in the absence or presence of 10 mg/kg IgG or GDF15 antibody. Longitudinal or endpoint measurements of tumor volume (**A**), tumor weight (**B**), food intake (**C**), tumor-free body weight (**D**), fat mass (**E**), and tumor-free lean mass (**H**) were obtained as described in Methods. At sacrifice, eWAT and gastrocnemius were harvested and processed for H&E (**F**) or Laminin (**I**) staining, respectively (scale bar 100 µm). Subsequently, measurements of adipocyte cross-sectional area (**G**) (*n* = 4 mice) and myocyte (**J**) cross-sectional area (*n* = 4 for KP, *n* = 3 for KPL tumor bearing mice without treatment, and *n* = 4 for KPL tumor bearing mice treated with IgG or GDF15 antibody) were obtained. Data are shown as mean ± SEM of the actual measurements (**A–D**, **G** and **J**) or relative to their day 0 values (**D–E**, **H**). *P* was calculated using 1-way (**B–D**, **G**, **H**, **J**) or 2-way (**A** and **E**) ANOVA followed by Dunnett's multiple-comparison test for significant differences from the KPL + GDF15 antibody cohorts. n.s. not significant. Source data are provided as a Source Data file.

and processing of GDF15, the latter necessary for cachexia development in vivo. As expected, overexpression of the wild-type version of the tumor suppressor STK11/LKB1 led to a moderate suppression of tumor growth kinetics compared to control tumors (Fig. 8D), but animals were only sacrificed for cachexia evaluations once all tumors from all cohorts reached similar terminal sizes (Fig. 8D).

In parallel to the cell and tumor findings, mice bearing the STK11/LKB1-reconstituted H1437^STK11 tumors demonstrated human serum GDF15 concentrations that approached baseline levels (i.e. equivalent to mice bearing no tumor) in comparison to the elevated serum GDF15 concentrations observed in mice bearing parental 1437 tumors or 1437 tumors overexpressing control vectors (Fig. 8E). The decrease in serum human GDF15 was associated with an increase in food intake in H1437^STK11 tumor bearing mice compared to H1437^CTL tumor bearing mice to levels similar to PBS-injected mice (Fig. 8F). Mice bearing H1437^STK11 tumors displayed a preservation of body weight (Fig. 8G), fat mass (Fig. 8H), and lean mass (Fig. 8I) when compared to mice bearing

parental 1437 or 1437^ctrl tumors. Quantitative RT-PCR analysis of the cell lines demonstrated that the reconstitution of STK11/LKB1 profoundly limited the expression of the ISR-related molecules *GDF15* and *ATF4* compared to expression in control cells (Fig. 8J).

Since reconstitution of wild-type STK11/LKB1 decreased transcription and protein expression of GDF15, we could not assess the role of STK11/LKB1 in inducing the processing of the proprotein form of GDF15. To clarify this question, we overexpressed the FLAG-tagged version of the pro form of GDF15 into H1437^CTL and H1437^STK11 cell lines (Fig. 8K). Immunoblot analysis using anti-FLAG antibody revealed that the lack of *STK11/LKB1* allowed for processing of the proprotein form of GDF15 into the mature protein in cells that was secreted into the medium. Whereas, overexpression of this construct into the wild-type STK11/LKB1 reconstituted cells only showed the proprotein form of GDF15 in the cells without the processed/mature form found in the cells or secreted into the medium. Furthermore, mature GDF15 protein is similarly lost in this system (medium and cells) with silencing of

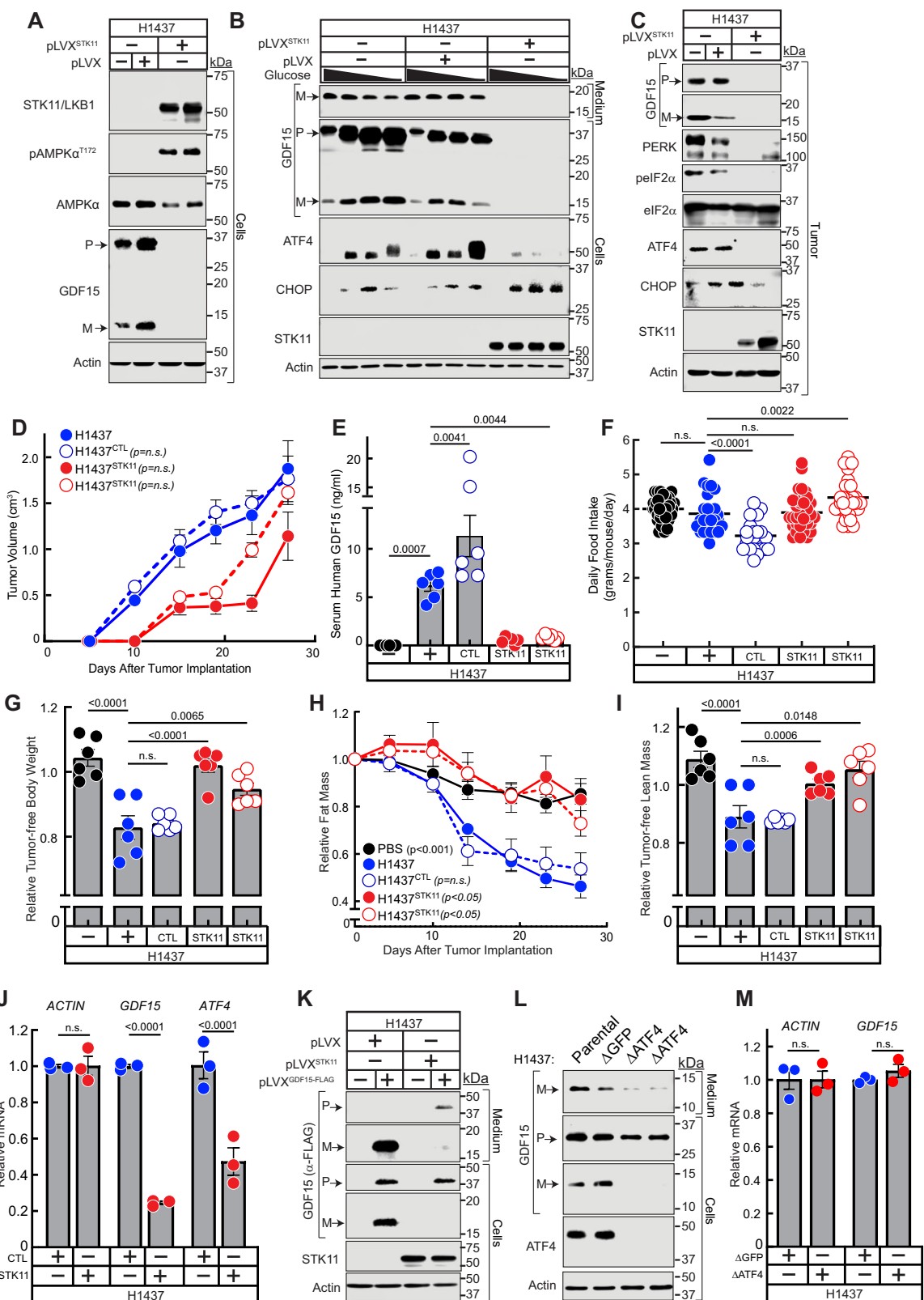

ATF4 in H1437 cells (Fig. 8L). The effect on eliminating mature GDF15 levels in cells or medium was not correlated with any change in *GDF15* RNA expression with the absence of ATF4 protein (Fig. 8M). Ultimately, findings from these experiments continue to highlight a dependence of *STK11/LKB1*-mutated NSCLC tumors on GDF15 protein expression and processing for the induction of cachexia wasting. Reciprocally, the studies highlight the role of the stress response molecules STK11/LKB1

and ATF4 in promoting GDF15 processing into a mature, active form of the protein.

## Discussion

This study was undertaken to better characterize cachexia induction when non-small cell lung cancers lose STK11/LKB1 functional capacity. Our previous study demonstrated that tumor cells with *STK11/LKB1*

**Fig. 8 | Effect of wild-type STK11/LKB1 reconstitution on GDF15 processing and cachexia induction. A, B** Expression and secretion of GDF15 from patient-derived NSCLC cell line reconstituted with wild-type *STK11/LKB1*. Parental H1437, H1437^CTL, or H1437^STK11 cells were harvested (**A, J**) or incubated with medium D in the presence of 0.01–10 mM glucose followed by harvesting (**B**). **D-I** Effect of STK11/LKB1 reconstitution on cachexia phenotype. Chow-fed 15-week-old NOD/SCID male mice (*n* = 6 per group) were injected s.c. with 200 μl PBS in the absence or presence of 1 × 10^7 cells from parental H1437, H1437^CTL, H1437^STK11, or another H1437^STK11 monoclonal line as described in Methods. Longitudinal or endpoint measurements of tumor volume (**D**), food intake (**F**), tumor-free body weight (**G**), fat mass (**H**), or tumor-free lean mass (**I**) were obtained as described in Methods. Tumors at sacrifice were processed for immunoblot analysis with the indicated antibody as described in Methods (**C**). At sacrifice, serum was subjected to the human GDF15 ELISA as described in Methods (**E**). H1437^CTL, or H1437^STK11 cells stably expressed

control vector or human GDF15-3xFlag protein, or parental H1437, H1437^ΔGFP or H1437^ΔATF4 were harvested (**K, M**) or incubated with medium D in the presence of 10 mM or 0.1 mM glucose followed by harvesting (**L**). Medium and cell lysate were subjected to immunoblot analysis with the indicated antibody or quantitative RT-PCR with indicated primer, respectively as described in Methods. Data are shown as mean ± SEM of the actual measurements (**D–F**) or relative to day 0 values (**G–I**) or values relative to control group (**J, M**). Samples for quantitative RT-PCR analysis of gene expression were from 3 biological replicates per group (**J, M**). *P* was calculated using 1-way (**E–G, I–J, M**) or 2-way (**D, H**) ANOVA followed by Dunnett's multiple-comparison test for significant differences from the H1437 parental cohort. n.s. not significant; M = GDF15 mature protein; P = GDF15 proprotein. The samples derive from the same experiment but different gels for different antibodies (**A–C, K, L**). Experiment was repeated at least 3 times with similar results (**A–C**). Source data are provided as a Source Data file.

loss-of-function variants promoted cachexia associated weight, adipose, and muscle loss when transplanted into mice[1]. Tumor loss-of-function variants in *STK11/LKB1* were also associated with clinical weight loss at cancer diagnosis in advanced NSCLC patients. We, therefore, initiated a candidate approach to identify downstream molecules with metabolic associations that were enriched when tumor STK11/LKB1 function was lost. One such candidate was GDF15, a TGFβ family member and anorexia modulator[3–5]. GDF15 is induced in host tissues of patients in settings of stress from nutrient deprivation and/ or cytotoxic chemotherapy administration[11–13,16]. In this manuscript we showed that mice bearing patient-derived human *STK11/LKB1* variant NSCLCs with cachexia-inducing capacity displayed increased tumor cell expression and elevated serum concentrations of tumor cell-derived GDF15 compared to mice bearing wild-type *STK11/LKB1* NSCLCs. To further establish that the tumor, and not host, was the primary source of GDF15 relevant to *STK11/LKB1*-mutated NSCLC cachexia, we silenced tumor-cell *GDF15* transcript specifically in multiple patient-derived lines before transplanting them into immunodeficient animals. These mice, at baseline demonstrating a spectrum of systemic GDF15 levels driven by parental NSCLC biology, all were significantly protected from cachexia development with the *GDF15* tumor-cell specific silencing, even in mice bearing tumors associated with low serum GDF15 levels. Critically, our findings were not affected by immunocompetent vs immunodeficient host backgrounds, male vs female status, or subcutaneous vs orthotopic anatomic location of transplantation. Reciprocally, reconstitution of wild-type *STK11/LKB11* in a patient-derived NSCLC line containing a *STK11/LKB1* gene deletion decreased the integrated stress response (ISR), manifested by increased PERK, eIF2α phosphorylation, and increased ATF4 levels, all culminating in a decrease in production (ATF4 independent) and processing (ATF4 dependent) of GDF15 proprotein and the secretion of its mature form. The net effect of this suppression of the ISR was a block in cachexia development. From a therapeutic perspective, GDF15 neutralizing antibodies are currently being tested in randomized Phase 3 trials in cancer patients to limit cachexia. We used similar antibodies in multiple animal models bearing *STK11/LKB1*-mutated cancers and also observed a suppression of GDF15 actions to abrogate cachexia. In aggregate, this data suggests that tumor-derived GDF15 is a conduit through which STK11/LKB1 loss-of-function in NSCLC supports the development of cachexia-associated adipose, muscle, and body weight loss.

Historically, the field of cachexia has focused on host tissue as the ultimate source of GDF15-associated anorexia and wasting, particularly under the stress of cytotoxic chemotherapy during tumor-directed therapy[16]. The association between GDF15 and cachexia is certainly not all that surprising, as multiple groups now have linked cytotoxic chemotherapy to the induction of host GDF15 to promote chemotherapy-associated anorexia/cachexia[17]. With cachexia incidence still high across multiple primary cancers despite reduced use of cytotoxic chemotherapies, it became apparent that tumor cells themselves

could be a contributor to systemic GDF15 levels than previously appreciated. Recently, the Tracer-X study identified an association between serum GDF15 levels and cachexia outcomes in NSCLC patients[18]. However, since that group did not identify an association between tumor expression and circulating concentrations of GDF15, they concluded the source of circulating GDF15 was likely derived from the host. Having accumulated 8 patient-derived *STK11/LKB1*-mutant NSCLC lines and engineered multiple pairs of isogenic lines with or without functional STK11/LKB1, we were able to test the relevance of GDF15 and its cellular (tumor versus host) sources in regulating cachexia development. These studies demonstrated that GDF15 secreted from *STK11/LKB1*-mutant tumors and not host tissue was critical for cachexia development, understanding the limitation of our immunodeficient model is missing part of the immune system. Using either neutralizing antibody or silencing of *GDF15* in tumor cells, tumor cell-derived GDF15 was relevant to wasting development and abrogation without affecting tumor progression, regardless of the systemic levels of GDF15. Interestingly, even in models with GDF15 serum concentrations at or below non-cachexia-inducing control models, we could still abrogate cachexia development with GDF15 neutralizing antibodies or silencing of tumor *GDF15*. There could be several explanations for this finding: 1) The ELISA we used could not detect small, but still clinically meaningful GDF15 levels, 2) At very low concentrations, GDF15's effects were peripheral and not central, 3) Local increases in GDF15 could act on peripheral neurons to effectuate changes in wasting, and/or 4) Neutralization of GDF15 has effects on another molecule that was relevant to wasting in mice bearing STK11/LKB1-mutated tumors. In these model conditions, it will be critical to determine whether GDF15 is acting peripherally or centrally to promote wasting. We plan to use syngeneic model systems to characterize *STK11/LKB1*-mutant tumor-derived GDF15 central and peripheral effects leveraging tissue-specific knockout models of the GDF15 receptor, GFRAL. We will simultaneously conduct single cell studies on tumor and host tissues to provide information on potential peripheral GDF15 signaling mechanisms associated with cachexia development.

Any in vivo studies of GDF15 would be remiss without a thorough understanding of alterations in food intake with time, tumor development, and cachexia induction. We conducted longitudinal evaluations of food intake with all our studies. Interestingly, some cohorts showed a consistent and expected change in food intake, i.e. a decrease in food intake, associated with elevated GDF15 serum concentrations when comparing experimental groups vs controls. Other studies did not show an expected increase in food intake with tumor silencing of *GDF15* even though there was rescue of the cachexia phenotype. We could explain these findings with one of several hypotheses. With smaller increases in GDF15 serum concentrations, there might not have been enough of a signal to centrally regulate food intake but still enough signal to promote wasting peripherally at the level of adipose and muscle. Alternatively, it could be that our studies were not sensitive enough to pick up subtle changes in mouse food

intake. A more thorough evaluation of food intake with use of metabolic cages will also better demonstrate GDF15 sites of action with results interrupted understanding singly-housed mice itself might alter food intake. It is important to note that any peripheral effect on wasting by GDF15 could be translated from its actions directly on host tissues or indirectly through a central mechanism.

However, we believe that other possible explanations for the inconsistency in food intake alterations associated with changes in circulating GDF15 concentrations may be more relevant to our studies and models of cancer cachexia. We specifically did not observe significant changes in food intake in our cancer-driven murine cachexia models. These findings are consistent with the work of another group who also studied GDF15 dependent cachexia in the context of malignancy in preclinical models[23]. In epidemiologic/clinical studies, cancer patients with cachexia syndrome and elevated serum GDF15 concentrations also did not demonstrate a decrease in appetite[20]. These observations, however, are fundamentally different from those taken of murine models in which recombinant GDF15 is administered to induce wasting, with mice displaying much less food intake over time[23]. In our studies with cancer-induced cachexia, we previously showed that food intake can be influenced by multiple factors, including a body's compensatory system composed of circulating leptin alterations as animals lose fat mass[25]. With cachexia-induced fat loss, there is less leptin in circulation in an attempt to promote food intake to limit the effects of starvation. Only by suppressing the leptin compensatory system genetically (*ob/ob* and *db/db* mice) did we demonstrate a perpetual food intake change when mice were administered a cachexia factor[25]. In our *STK11/LKB1*-mutated NSCLC cachexia studies with intact GDF15, we also observed a decrease in serum leptin concentrations as mice undergo adipose wasting and body weight loss. This compensatory reduction in leptin concentrations likely masked any food intake-dependent changes by GDF15. Finally, other groups have shown that exogenous administration of GDF15 promotes body weight loss through food intake dependent and independent mechanisms[22]. In mice with MASH modeled pathologically, GDF15 administration caused further body weight loss compared with pair-fed mice by increasing energy expenditure in muscle with sarcolipin induction[22]. In another study, GDF15 neutralization could ameliorate muscle atrophy and body weight loss with an associated decrease in gastrocnemius muscle sarcolipin in a mouse model of mitochondrial myopathy[33]. These same observations are made with our pathologically-driven cancer cachexia models – an increase in muscle expression of sarcolipin associates with GDF15-dependent body weight loss. Therefore, wasting despite the lack of food intake changes observed with GDF15 induction and subsequent increases in serum concentrations in cancer cachexia models is possibly explained by host leptin compensatory mechanisms and GDF15 food intake independent energy expenditure mechanisms.

On a parallel note, with our model systems, we were able to fill in knowledge gaps with regard to tumor growth kinetics as a function of cachexia induction and vice versa. When we utilized the GDF15 neutralizing antibody in our cachexia studies, we were uniformly able to suppress wasting without seeing a consistent change in tumor growth kinetics, suggesting that growth of STK11/LKB1-mutated NSCLC cachexia-inducing tumors was not dependent on wasting by-products to profit through proliferation. Therefore, in our studies of cachexia, the growth of tumors with at least *STK11/LKB1* loss-of-function variants is not sensitive to wasting output. This cannot be extrapolated to tumors that grow secondary to other mutations or genetic drivers and is an area of active research. Finally, our studies also suggest a divergence in mechanisms downstream of *STK11/LKB1* loss of function with respect to regulation of tumor cell proliferation and cachexia wasting. Reconstitution of wild-type *STK11/LKB1* in H1437 tumor cells suppressed both tumor growth and cachexia induction. However, GDF15 neutralization or silencing in multiple lines only abrogated cachexia and not tumor growth suggesting a divergence of the *STK11/LKB1*-mutated cancer progression and cachexia pathways in tumor cells.

Collectively, our work in this paper highlights the dependence of tumor cells with *STK11/LKB1* loss-of-function variants on increasing levels of GDF15 to promote cachexia, through either the increase of transcript, processing, and/or secretion of the protein. In our model systems, the tumor cells and not the host tissue are the source of GDF15 and its associated wasting. Both silencing of *GDF15* in tumor cells or use of a GDF15 neutralizing antibody systemically in multiple patient-derived and mouse-derived lines suppress *STK11/LKB1*-mutated NSCLC cachexia regardless of circulating GDF15 concentrations. This GDF15 action in regulating STK11/LKB1-associated cachexia was associated with an exaggerated tumor cell ISR. Considering the preclinical findings linking GDF15 activation to *STK11/LKB1*-mutated NSCLC, clinical translation of these findings will require testing of the ~15% NSCLC patients who possess *STK11/LKB1* variants to determine if the cachexia benefit from GDF15 inhibitors can similarly be observed across a wide range of systemic/circulating GDF15 levels.

## Methods

### Study approval
Animal studies were approved through the University of Texas Southwestern Medical Center's Institutional Animal Care and Use Committee (protocol 2015-100994).

### Materials
Detailed material information is listed in Supplementary Table 1.

### Buffer and culture medium
Buffer A contained 10 mM Tris-HCl (pH 6.8), 100 mM NaCl, 1% (w/v) SDS, 1 mM EDTA, 1 mM EGTA, Phosphatase Inhibitor Cocktail Set I and Set II, and Protease Inhibitor Cocktail. Medium A was RPMI 1640 supplemented with 5% (v/v) fetal bovine serum (FBS), 100 units/ml penicillin, and 100 µg/ml of streptomycin. Medium B was DMEM high glucose supplemented with 100 units/ml penicillin and 100 µg/ml of streptomycin. Medium C was medium B supplemented with 10% (v/v) FBS. Medium D was RPMI 1640 without glucose supplemented with 0.5% FBS, 100 units/ml penicillin, and 100 µg/ml of streptomycin.

### Cell culture
Stock cultures of all human NSCLC cell lines (kind gift from John Minna), KP, KPL cell line (kind gift from Esra Akbay), and HEK293T cells (ATCC) were maintained in monolayer culture at 37 °C in 5% $CO_2$ in medium A or medium C, respectively. Each cell line was propagated, aliquoted, and stored under liquid nitrogen. Aliquots of these cell lines were passaged for less than 4 weeks to minimize genomic instability prior to injection into mice. Every six months, the cell lines were tested for *Mycoplasma* contamination using the MycoStrip *Mycoplasma* Detection Kit.

### Generation of gene knockout CRISPR/Cas9 lentivirus
A single guide RNA (sgRNA) directed to *GFP* as non-targeting control (ΔGFP) or to human *GDF15* (ΔGDF15) were cloned into the *Bsm*BI (Thermo, #ER0451) site of lentiCRISPRv2-puro vector (AddGene, #52961) using the following pairs of annealed oligonucleotides obtained from IDT for *GFP* (forward 5′-CACCGGTGAACCGCATCG AGCTGA-3′; reverse 5′-AAACTCAGCTCGATGCGGTTCACC-3′), or human *GDF15* (forward 5′-CACCGGGACGTGACACGACCGCTG-3′; reverse 5′-AAAC CAGCGGTCGTGTCACGTCCC-3′), or human *ATF4* (forward 5′-CACCGATCCACAGCCAGCCATTCGG-3′; reverse 5′- AAACC CGAATGGCTGGCTGTGGATC-3′). These constructs were transformed into stable competent *E. Coli* (NEB, #C3040H) and constructs were purified using the manufacture's instruction of the MIDI-Prep system (Macherey-Nagel, #740410.10). The cloned nucleotides were confirmed by sanger sequencing. For lentiviral production, HEK293T cells

were seeded in 100 mm dishes at ~1×10⁶ cells in 10 ml medium D on day 0. On day 2 (~70–80% confluence), medium was then removed and replaced with 8 ml medium C supplemented with an additional 2 mM glutamine, and 15 µl X-tremeGENE HP DNA Transfection Reagent, 2.5 µg of the indicated sgRNA-lentiCRISPRv2, 1 µg of pMD2.G (AddGene, #12259), and 1.5 µg of psPAX2 (AddGene, #12260) mixed in 0.8 ml Opti-MEM medium (Thermo, #31985070). On day 3, medium was removed and treated with 11 ml medium B with 30% FBS supplemented with an additional 2 mM glutamine. On day 4, 5, and 6, virus-containing medium was collected and exchanged with 11 ml of fresh medium B with 30% FBS supplemented with an additional 2 mM glutamine. The collected virus-containing medium was filtered through a 0.45 µm syringe filter unit (Millipore Sigma) and stored at −80 °C for future use.

## Generation of wild-type STK11/LKB1 and GDF15-3x-flag overexpression lentivirus

Complimentary DNA library was derived from the reverse transcription (Thermo, #N8080234) of extracted RNA from the H1792 line as described by the manufacturer's directions. Human *STK11* (NM_000455.5) complimentary DNA containing the EcoRI and BamHI restriction sites at the 5′ and 3′ ends, respectively, was amplified using the following pairs of primers (forward 5′-TATTTCCGGTGAATTCatg gaggtggtggacc-3′; reverse 5′-GAGAGGGGCGGGATCCtcactgctgcttgc ag-3′). This fragment was cloned into the pLVX-EF1α-IRES-Puro vector (Clontech, #631988) by first treating with the EcoRI-HF (NEB, #R3101S) and BamH1-HF (NEB, #R3136S) restriction enzymes followed by In-Fusion Snap Assembly (Takara, #638947) per the manufactures' directions. Human GDF15 (NM_004864.2) ORF was purchased from UT Southwestern McDermott Center for Human Genetics, that was a clone from the Ulitmate ORF Lite human cDNA collection (Life Technologies). GDF15-3x-Flag complimentary DNA containing the XbaI and BamHI restriction sites at the 5′ and 3′ ends, respectively, was amplified using the following pairs of primers (forward 5′- CGAGACTAGTTC TAGAGCCACCatgcccgggcaagaac-3′; reverse 5′- GAGAGGGGCGGGAT CCctacttgtcatcgtcatccttgtaatcgatatcatgatctttataatcaccgtcatggtctttgta gtctatgcagtggcagtc-3′). This fragment was cloned into the pLVX-EF1α-IRES-Hygro vector (kindly gifted by Shimeng Xu, UTSW Medical Center) by first treating with the XbaI (NEB, #R0145S) and BamH1-HF (NEB, #R3136S) restriction enzymes followed by In-Fusion Snap Assembly (Takara, #638947) per the manufactures' directions. These constructs were transformed into stable competent *E. Coli* (NEB, #C3040H) and constructs were purified using the manufacture's instruction of the MIDI-Prep system (Macherey-Nagel, #740410.10). The cloned nucleotides were confirmed by sanger sequencing. For lentiviral production, HEK293T cells were seeded in 100 mm dishes at ~1×10⁶ cells in 10 ml medium D on day 0. On day 2 (~70–80% confluence), medium was then removed and replaced with 8 ml medium C supplemented with 2 mM glutamine, and 15 µl X-tremeGENE HP DNA Transfection Reagent, 2.5 µg of the indicated pLVX-EF1α-IRES-Puro construct, 1 µg of pMD2.G (AddGene, #12259), and 1.5 µg of psPAX2 (AddGene, #12260) mixed in 0.8 ml Opti-MEM medium (Thermo, # 31985070). On day 3, medium was removed and treated with 11 ml medium B with 30% FBS supplemented with an additional 2 mM glutamine. On day 4, 5, and 6, virus-containing medium was collected and exchanged with 11 ml of fresh medium B with 30% FBS supplemented with an additional 2 mM glutamine. The collected virus-containing medium was filtered through a 0.45 µm syringe filter unit (Millipore Sigma) and stored at −80 °C for future use.

## Generation of knockout or overexpression cell lines

Monoclonal parental, empty vector overexpressed, or wild-type STK11/LKB1 overexpressed H1437 NSCLC lines were seeded in 6-well dishes at ~1×10⁵ cell density in 2 ml of medium A. After 24 h the medium was replaced with fresh medium A (non-transduced control) or 1 ml

medium A and 1 ml of the indicated virus-containing medium, supplemented with 8 µg/ml polybrene reagent (Millipore Sigma, #TR-1003). After 24 h, the medium was removed and replaced with 2 ml of medium A. The next day, the medium was removed and replaced with fresh 2 ml medium A containing 1 µg/ml puromycin or 300 µg/ml hygromycin, and puromycin- or hygromycin-supplemented medium A was refreshed every 2 days until complete cell death was observed in the non-transduced control. For knock out or wide-type STK11/LKB1 overexpression, limiting dilution of these puromycin-selected cells was performed in puromycin-containing medium to produce single colonies of transduced cells, and these monoclonal knockout lines were subsequently propagated and aliquoted for storage under liquid nitrogen or further propagated for experiments described herein.

## Cancer cachexia animal models

All animal studies were conducted under an Institutional Animal Care and Use Committee (IACUC)-approved protocol at UT Southwestern Medical Center (Dallas, Texas). NOD.CB17-Prkdc<scid>/J (*NOD/SCID*) and C57BL/6J mice were obtained from The Jackson Laboratory at approximately 9 weeks of age. All mice were allowed to acclimate in UT Southwestern animal facilities for at least 2 weeks. Animals were kept in a temperature-controlled facility (at ~22 °C) with a 12 h light/dark cycle and were fed regular chow diet.

On day 0 of the study, *NOD/SCID* or C57BL/6J mice underwent baseline assessments of body weight (digital Ohaus scale) and both lean and fat mass components of body composition using ECHO MRI (ECHO Medical Systems).

For subcutaneous model, 200 µl PBS in the absence or presence of either 1×10⁷ human or 1×10⁶ mouse NSCLC cells (parental or genetically engineered lines) were injected into the right flank of the *NOD/SCID* or C57BL/6J mice, respectively.

For orthotopic model, the procedure for surgical orthotopic transplantation of lung cancer cells has been published previously[27–29]. Briefly, a 1-cm incision was made through the skin and underlying subcutaneous fat of mice anesthetized under continuously metered 2% isoflurane. PBS or 1×10⁶ tumor cells (H1573 GFP-KO or GDF15-KO) suspended in 30 µl of 25% growth factor reduced Matrigel in dye-free RPMI was injected via a 30-gauge needle through the intercostal muscles directly into left lung, respectively. Skin was stapled together using 9-mm staples. Carboprofen 5 mg/kg was administered intraperitoneally after surgery and the following two days for pain control. Mice were monitored until fully recovered from anesthesia. After 12 days' recovery, staples were removed, and mice were monitored as below.

Food intake measurements were conducted daily as described previously, including the method for calculation of average daily food intake[25]. Every 2–4 days, mice underwent longitudinal measurements of body weight, both lean and fat mass components of body composition, and tumor size by caliper (VWR) measurements of length, width, and breadth.

UT Southwestern IACUC policy for single subcutaneous tumor requires euthanasia of animals when the diameter of any one tumor is greater than 2 cm in a mouse. The timing of animal sacrifice for tumor-injected cohorts was a function of imminent animal death (expected within 12 h), tumor approaching 2.0 cm diameter, or cohorts of animal reaching about 50% fat loss; PBS cohorts were sacrificed concurrently with matched tumor cohorts. At sacrifice, mice were euthanized as recommended by the IACUC with use of a CO₂ chamber. Tumor weights were measured and collected and snap frozen in liquid nitrogen and placed in long-term storage at −80 °C (for future analysis). Whole blood at sacrifice was obtained through cardiac puncture and serum was obtained by subjecting the whole blood to centrifugation at 960×g at 4 °C for 10 min followed by collection of the supernatant.

Longitudinal values for tumor volume were calculated as half the product of caliper measurements of length, width, and breadth. Longitudinal and terminal values for body weight, lean mass, and fat mass were calculated relative to both the matched animal day zero value and mean value of the matched PBS cohort. At sacrifice, relative tumor-free body weight and lean mass were calculated by subtracting the matched animal tumor weight at sacrifice prior to normalization.

## Serum protein concentration determination

Human GDF15 in serum was measured using a human GDF15 Quantikine ELISA kit (R&D Systems, #DGD150,) following the manufacturer's instructions. Mouse GDF15 in serum was measured using a mouse/rat GDF15 Quantikine ELISA kit (R&D Systems, #MGD150) following the manufacturer's instructions. Mouse Leptin in serum was measured using a mouse ELISA kit (Crystal Chem, #90030) following the manufacturer's instructions.

## Immunoblot analysis

Human NSCLC cell lines cells were seeded in 6-well dishes at ~$4 \times 10^5$ cell density in 2 ml of medium A. After 24 h cells were washed with 2 ml DPBS, then treated with Medium A or Medium D in the absence or presence of different concentrations of glucose as indicated. After 16 h, media were harvested and spun with $800 \times g$ for 2 min and the supernatant was concentrated 15-fold using a 3 kDa filter (Amicon Ultra) per manufacturer's instructions to obtain the concentrate. Cells were washed with 2 ml DPBS, then cells were treated with 170 µl of buffer A then subjected to a plate shaker for 2 min at 700 RPM followed by harvesting of the cell lysate which was subjected to 10 s of sonication (Cole-Parmer, #EW-04714-50) at 50% amplitude. The concentrated media and cell lysate were then used for immunoblot analysis.

Tumor from mouse studies were thawed on ice and ~50–60 mg of tumor tissue was placed in a 2 ml microtubes with 1 ml of Buffer A and homogenized with a Bead Ruptor Elite apparatus (Omni International) for three cycles of 20 s at 4 m/s, followed by 10 s of sonication (Cole-Parmer, #EW-04714-50) at 50% amplitude. The homogenized sample was immediately subjected to centrifugation at 10 °C for 10 min at $20,000 \times g$. The supernatant was separated from the pellet and collected. The tissue lysate were then used for immunoblot analysis.

Protein concentrations of cell lysate, or mouse tumor lysate were measured using a Pierce™ bicinchoninic acid kit and then mixed with 5X loading buffer (250 mM Tris-HCl (pH 6.8), 10% sodium dodecyl sulfate, 25% glycerol, 5% β-mercaptoethanol, and 0.2% bromophenol blue) heated at 95 °C for 5 min, and subjected to 10% or 15% SDS/PAGE (8 µg/lane for cell/tumor lysate, or volume of cell medium normalized to cell protein concentration). The electrophoresed proteins were transferred to nitrocellulose membranes using the Bio-Rad Trans Blot Turbo system, followed by incubation in blocking buffer containing 5% (w/v) non-fat powdered milk in PBS-T. Blocked membranes were washed briefly in PBS-T, followed by primary staining with indicated primary antibodies diluted in primary antibody solution-Can Get Signal (TOYOBO). Membranes were again washed in PBS-T, followed by secondary staining with donkey anti-mouse or rabbit IgG conjugated to horseradish peroxidase diluted in the blocking buffer. Membranes were again washed in PBS-T, and bound antibodies were visualized by brief incubation in chemiluminescent substrate solution followed by imaging using an Odyssey FC Imager Dual-mode Imaging System (2-min integration time). Immunoblot images were analyzed using Image Studio software (LICOR).

**Real-time PCR analysis of gene expression.** Extraction of total RNA with RNA-STAT-60 reagent from cells, tumors, and muscles, subsequent quantitative reverse transcriptase PCR assays for mRNA levels of the indicated gene products were conducted as previously described (1). The sequences of primers (obtained from Integrated DNA Technologies) used in these studies are as follows:

| Gene product | Origin | Symbol | Forward | Reverse |
|---|---|---|---|---|
| β-actin | Human | *ACTB* | CACCATTGGCA ATGAGCGGTTC | AGGTCTTTGCG GATGTCCACGT |
| | Mouse | *Actb* | CCGTGAAAAGA TGACCCAGATC | CACAGCCTGG ATGGCTACGT |
| Growth differentiation factor 15 (GDF15) | Human | *GDF15* | CAACCAGAGCT GGGAAGATTCG | CCCGAGAGATA CGCAGGTGCA |
| | Mouse | *Gdf15* | CTCTCAACTGA GGTTCCTGC | CCAATCTCACC TCTGGACTG |
| F-box protein 32 (Fbxo32) | Mouse | *Fbxo32* | GAGGGCCATTG ACTTTGGGAC | CTCCTTCTTCAT TGGTGTTCTTCT |
| Tripartite motif containing 63 (Trim63) | Mouse | *Trim63* | GAGGGCCATTG ACTTTGGGAC | CTCCTTCTTCAT TGGTGTTCTTCT |
| Activating transcription factor 4 (ATF4) | Human | *ATF4* | CTCCAACAACAG CAAGGAGGA | TACCCAACAGG GCATCCAAG |
| Sarcolipin | Mouse | *Sln* | TGTGCCCCTGC TCCTCTTC | TGATTGCACAC CAAGGCTTG |

**Microscopic analysis of mouse adipose, muscle tissue and lung tumor area.** Epididymal white fat was collected, and immersion fixed in 10% neutral-buffered formalin. All samples were paraffin processed by members of UT Southwestern's Histo Pathology Core using standard histologic technique. Resulting embeds were sectioned at 5 µm, and produced slides were stained by regressive methodology utilizing Leica Selectech reagents (Hematoxylin 560, Define, Blue Buffer, and Eosin-Y 515) on a Tissue-Tek Prisma Plus robotic stainer (Sakura-USA). From the stained slides, microscopy fields were imaged on a BZ-X scanner (Keyence) at 20× magnification and adipocyte area was measured using ImageJ software Adiposoft plugin (Fiji) for three fields from each slide (250–350 adipocytes per slide). Each group contains the quantification of slides prepared from indicated number of mice.

Cryosections of whole Gastrocnemius muscle were prepared by UT Southwestern's Histo Pathology Core and preserved at −80 °C freezer. When staining, slides were brought to room temperature and surrounded with liquid blocker. Slides were blocked with blocking buffer (5% goat serum in PBS) in a wet chamber for at least 30 min at room temperature. Slides were incubated with Laminin antibody (1:1000 in blocking buffer) overnight at 4 °C. The next day, slides were washed with PBS 3 times. Slides were incubated for at least 1 h with secondary antibody (Anti-rabbit-Alexa Fluor 488 was diluted1:1000 in blocking buffer). Slides were washed with PBS 4 times. Slides are covered with immunostaining mounting media and cover slides. Fluorescent images were captured using the Keyence BZ-X700 microscope. Three fields from each slide were imaged and myofiber cross-sectional areas were measured using Ilastik (EMBL Heidelberg) and ImageJ (Fiji) software[34].

Paraffin embedded blocks of lungs were made in UTSW Tissue Management Shared Resource Facility. H&E images of left lungs were scanned with Hamamatsu NanoZoomer 2.0 HT in UTSW Whole Brain Microscopy Facility. Images visualization, tumor and lung areas were calculated using Hamamatsu NDPView2 software. Analysis of the percentage of tumor area was performed using GraphPad Prism 10.

## Forelimb grip strength test

Grip strength was measured using a BIO-GS4 grip strength test meter (Bioseb Instruments, France). Mice were placed on a metal grid connected to the meter and slowly pulled by the tail. The meter records the maximum grip strength (gram, and was converted to newton (N) by multiplying 0.0098). The measurement was repeated five times and

averaged to determine the grip strength of each mouse. Relative grip strength is normalized with day 0 body weight and shown in Supplementary Data 1.

## Statistics & reproducibility

For subcutaneous models, each group includes 6 mice; for orthotopic model, each group includes 9 mice based on previously published literature. Mice data points were excluded from the results when there were no tumor forming or mice died unexpected before data collection. The experiments were not randomized, and the investigators were not blinded to allocation during experiments and outcome assessment.

Details of statistical analysis for each experiment can be found in the respective figure legend. Data is presented as mean ± SEM, dot plots ± SEM, dot plots with bars ± SEM, or histogram. For experimental designs with two conditions, unpaired parametric $t$ tests were conducted to investigate significant differences following tests to verify normality and equal variance of both groups. For experimental designs with greater than two conditions, one- or two-way analysis of variance (ANOVA) tests were conducted to examine if there were significant differences in outcomes among groups. For the case of multiple pairwise comparisons, Dunnett's multiple comparison post-test was used. All statistical analyses were computed using GraphPad Prism 10 software.

## Reporting summary

Further information on research design is available in the Nature Portfolio Reporting Summary linked to this article.

## Data availability

Source data are provided with this paper.

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

## Acknowledgements

We thank Michael Brown, Joseph Goldstein, Shawn Burgess, and Jay Horton for their suggestions and review. We also thank Bei Zhang (Pfizer), Mingjian Lu (Pfizer), Junjie Li (Pfizer), Ja Young Kim-Mueller (Pfizer) and Danna Breen (Pfizer) for their contributions. We thank Natalie Lopez and Min Ding for their excellent technical assistance. This work was supported by grants from the Burroughs Wellcome Fund Career Awards for Medical Scientists (1019692, R.E.I.); American Cancer Society Grant (133889-RSG-19-195-01-TBE, R.E.I.); Cancer Prevention and Research Institute of Texas (RP230140 (R.E.I./P.I.) and RP210041 (E.A.A.)); and National Institutes of Health grants (1R01CA266900 (R.E.I./P.I.), 1R01CA258684 (J.K.), 1R01DK128166 (R.E.I./P.I.), 1P30DK127984 (R.E.I.), P30CA142543 (J.D.M., J.K., R.E.I.), P50CA070907 (J.D.M., J.K., R.E.I.), and T32DK007745 (E.L.)).

## Author contributions

J.Y., T.G., A.G., N.W., E.L., J.K., B.A.R., P.I., and R.E.I. designed experiments. J.Y., T.G., A.G., N.W., E.L., D.Z., S.S., T.S., and J.M.S. conducted experiments and analyzed the data. Q.D., E.A.A., and J.D.M. provided reagents. B.M.E., Z.W., I.T., E.P., J.D.M., P.I. and R.E.I. analyzed data. J.Y. performed statistical analysis. J.Y., P.I., and R.E.I. wrote the manuscript. All authors reviewed and revised the manuscript.

## Competing interests

All authors acknowledge no conflicts of interest related to this work except R.E.I. who maintains collaborative cancer cachexia research projects with Pfizer, Inc. and J.D.M. who receives royalties from the National Institute of Health and University of Texas Southwestern Medical Center for distribution of human tumor cell lines.
