## [Transparent Peer Review file · Nature Communications]

Cancer Cachexia in STK11/LKB1-Mutated Non-Small Cell Lung Cancer is Dependent on Tumor-Secreted GDF15

Corresponding Author: Dr Rodney Infante

Version 0:

Reviewer comments:

Reviewer #1

(Remarks to the Author)

The current manuscript describes the impact of pharmacologic and genetic GDF15 targeting to prevent lung cancer (NSCLC)-induced cachexia. The importance of GDF15 in cancer cachexia was previously reported and anti-GDF15 clinical trials in NSCLC patient are currently ongoing. The main novelty of the manuscript looks the GDF15 secretion to depend from LKB1 mutations independently from anti-cancer therapies and that the mutations in the tumor may predict GDF15 production and cachexia development. The association between GDF15 production and LKB1 activity was previously reported (PMID: 38111379) in non-cancerous cells and several studies highlighted the link between GDF15 and AMPK. However, the mechanism linking the AMPK/LKB1 pathway modulation with GDF15 secretion is missing from both previous and current study. The research was mainly conducted implementing the use of subcutaneous xenotransplantation of human cells in immunodeficient mice, producing a limited translatable condition to human cachexia (acknowledged in the discussion). The authors thus implement the use of iso-/syn-graft in immunocompetent mice, obtaining non completely overlapping, although coherent data. Overall, the results are interesting and deserve attention, however, there are several points that require a careful revision either by providing new / newly elaborated data or by arguing.

Main issues:

The introduction is in my opinion too long and resembles partly a result and partly a discussion chapter, failing to provide the state of the art in the studied field in order to understand the following results. Considering that cancer cachexia does not develop in all cancer types, a brief description of the development of cancer cachexia in NSCLC could be added.

Animal sex is not specified. The authors should clarify which animal sex was chosen and explain the reason behind the choice.

In order to present homogeneous data for the different models adopted, the tumor and tissue weights at necropsy (at least muscle and adipose tissue) should be included for all the experiments. Considering that the lean mass assessment by ECHO-MRI cannot dissect the contribution of different tissues, and that the enlargement of a tissue could compensate for the reduction of another, the final tissue weights should be presented, even in a supplemental table.

Figure 1: measuring host contribution to GDF15 production in immunodeficient mice is in my opinion inappropriate, since the host response to the presence of the cancer is severely impaired. The authors describe the association between CC presence and high tumor GDF15 transcript and released peptide, however a recent observational clinical trial suggest the lack of association between cachexia and high GDF15 (PMID: 37115357). Please discuss.

Figure 2: tumor-free mice untreated, treated with IgG or anti-GDF15 are missing. Since the tumor-bearing mice treated with anti GDF15 gain weight/mass, I think it is important to show whether the anabolic response is present even in healthy controls despite having almost undetectable GDF15 levels.

Figure 3: Daily food intake is not reduced in H1573 tumor bearing mice showing high GDF15 levels and GDF15 blockade has no impact on food intake. The role of GDF15 in cachexia has been mainly described as an anorexigenic molecule, while it is not the case in the current model. I think the authors should better describe the mechanism underlying GDF15 action in the currently adopted model. Surprisingly, when injecting H2122 cells in mice, anorexia is present and silencing GDF15 protects from anorexia, although H2122 cells produce less GDF15 than H1573 (Figure 5). Again, when using H1437 cells (Figure 7 and S5), it looks like that the reduction of GDF15 release obtained by wt LKB1 reconstitution allows to counteract anorexia. Such scattered results are in overt contrast with the authors conclusion stating that 'Our studies further illustrated that when the GDF15 increases in serum concentrations were not as high, there was less effect on food intake' (Line 477). Some values of daily food intake seem to be very high, going above 7g/day/mouse (such as in figure 3F). Can the authors explain such high values? Food intake kinetic per group could be an interesting addition. Have the authors checked for food grinding? How many mice were present in each cage? Was any mouse excluded, sacrificed before the

established endpoint, producing a change in the feeding behavior of the remaining mice?

Figure 4: adipocyte and muscle fiber CSA in healthy controls are missing. 3 mice per cohort for histology looks too low and unlikely to provide such statistical differences. I wonder whether the average CSA from each mouse has been used as a single point or each adipocyte/fiber area has been considered as an independent point. The second option is incorrect. In the third panel of Figure 4E, there is an unexpected increase in mice injected with Δ GFP cells; this highlights a possible limitation of having only 3 mice per cohort. The same applies for Figure 5H, where there is an unexpected reduced CSA in Δ GFP.

Figure 6, panel B: is tumor volume at sacrifice statistically different? Tumor mass, anorexia, serum GDF15 and cachexia look directly correlated. In panel J from the same figure, it looks like that anti GDF15 initially promotes adipose mass accumulation in KPL-bearing mice, followed by a second phase of 'resistance/ineffectiveness'. A similar phenomenon seems to occur in mice bearing H1792 Δ STK11 tumors (Supplemental figure 4, panel E). Given the potential translational value of the study, the prophylactic nature of the design is not fully convincing, suggesting to perform a therapeutic study to show the impact of anti GDF15 in early cachectic mice (i.e. starting at day 15 or 20 from KPL cell injection). Similarly, a survival (or even a 'time to humane endpoint') study would strongly increase the translational value. Panel K: first time showing skeletal muscle weight; I would recommend including healthy controls. Similarly, relative tumor-free body weight and relative tumor-free lean mass for KPL + IgG/GDF15ab are missing.

Figure 7: very elegant gene reconstitution experiment, however, the reduced tumor growth rate in reconstituted LKB1 cells does not allow to conclude on the impact on cachexia, given that the same tumor mass developed in a different time frame likely induced a differential adaptation in the hosts, resulting in a different cachectic state. Panel D: the x axis of the graph has to be fixed.

I hardly understand the supplemental figure 5, where two experiments almost identical show contrasting or even opposite results. How can 2 weeks of age difference in adult mice produce such different results?

The authors state that GDF15 is relevant for muscle contractility. Considering that in this work GDF15 signaling is blocked, an assessment of muscle function (at least grip test and/or treadmill test, in case physiological muscle strength is not a lab commodity) should be included.

Minor issues:

Is it necessary to have 7 figures in the main text?

Fig4 legend, 'H1573 NSCLC lines allotransplanted into murine models': the cells are transplanted in mice, the model is obtained in the mouse transplanted with cells.

WB: overexposed and low quality of the results.

line 70, 'that were are not'

line 72, lack of isogenic models: the authors should clarify whether referring to the host, to the cancer cells, or both, given that there are several models of CC using syngeneic mice receiving almost monoclonal cancer cells originated from the same host animal strain.

Line 87, 'STK11/LKB1 functional variant': do the authors refer to an oncogenic mutant? I suggest to briefly explain in the text.

Line 146: please use the definition 'cachexia-inducing cell line' in spite of 'cachexia line'.

Line 221: despite having ECHO-MRI and CSA data, to state "adipose and muscle atrophy", the final tissue weights should be reported.

Line 237: missing comma after "food intake".

Line 482: "adipose and muscle tissues".

Lines 483-485: the authors propose the use of metabolic cages and single-housing to better study food intake alterations associated with changes in GDF15 circulating levels. While single-housing can be an interesting approach to isolate the behavior/contribution of each animal, changing animal housing numerosity can be an independent factor affecting the feeding behavior, possibly determining discrepancies with the observations presented in this study. The authors should mention this limitation in the proposed future approach.

Line 614: serum is obtained by collection, not removal of centrifuged whole blood supernatant.

Line 643: why sonication at this point and not after the disruption?

Line 859: the word "staining" is repeated.

Line 889: missing a point at the end of the sentence.

Line 927: "A-H".

Lines 928, 944, 959: please adjust group numerosity reported in the text (doesn't match with the related graphs).

Reviewer #2

(Remarks to the Author)

Reviewer #3

(Remarks to the Author)

This study from Yu et al. investigates the link between LKB1 loss and GDF15 in promoting cancer-associated cachexia in models of lung cancer. The authors demonstrate that loss of LKB1 in patient-derived lung tumor cells promotes an increase in GDF15 expression and a corresponding increase in circulating levels of GDF15. Treatment with an anti-GDF15 antibody alleviates features of cachexia in their cachexia inducing cell lines with mutation in LKB1/STK11. While the authors have demonstrated a clear link between loss of LKB1 and levels of GDF15 in their cancer cell lines, major reservations remain about the generalizability of this work due to the models of cachexia utilized in the manuscript. Additionally, there is little to no mechanistic insight provided into how loss of LKB1 regulates production of GDF15 both on the transcriptional and post

transcriptional level. Major revisions are requested to this manuscript.

Major comments:

The study relies predominantly on subcutaneous tumor transplantation of a large number of cells, with tumors often reaching extremely large sizes (<1cm³) before any meaningful weight loss is observed. In human patients, tumor burden is often poorly predictive of weight loss associated with cachexia, with small tumors capable of inducing cachexia. A main concern over the validity of the conclusions of this manuscript is that these subcutaneous tumor models are poorly reflective of human disease, and do not accurately represent the complex interplay between tumor genetics and the tumor microenvironment. In an autochthonous tumor model of LKB1 deficient NSCLC, where tumors develop in the native environment of the lung, levels of GDF15 were shown to be elevated as compared to LKB1 WT tumors but play no part in phenotypes associated with cachexia. The link between LKB1 loss and GDF15, and the role of GDF15 in promoting cachexia in these tumors, should be studied in models of NSCLC where tumors grow in the native environment in the lung, including genetically engineered mouse models and syngeneic transplant studies. Such studies will give insight as to whether the link between GDF15 and LKB1 loss is robust or an artifact of the model chosen.

There seems to be an intrinsic effect on neutralizing GDF15 even in tumors that do not have high levels of circulating GDF15 (in particular, Supplemental Figure 4). This suggests that treatment with anti-GDF15 antibody is driving the reversal of cachexia phenotypes rather than through blocking the actions of GDF15. To confirm an on target effect of GDF15 neutralization, the anti-GDF15 antibody studies should be complemented with treatment of tumors with anti-GFRAL antibody, the only known receptor of GDF15. Additionally, previous studies have indicated that the principal contributor of GDF15 mediated weight loss is from sympathetic output rather than changes in food intake. It would be insightful to see the effect of chemical sympathectomy by 6-OHDA treatment to observe whether this output is playing a role in cachexia in these models.

LKB1 regulates a host of downstream kinases including AMPK and the salt-inducible kinases (SIKs). Previous studies have indicated the tumor suppressive effects of LKB1 are dependent on the SIKs. Additionally, a CRTC2-CREB-SIK axis mediates the inflammatory potential of cells harboring loss of LKB1. It would be informative to see if increase in GDF15 expression is mediated through this same pathway or if other kinases downstream of LKB1 play a role.

The evidence linking lowering of GDF15 levels to STK11 reconstitution is clear in Figure 7, yet whether this lowering of GDF15 levels is primarily driving the improvement in weight loss is correlative. Studies utilizing GDF15 loss or neutralization should be performed to examine the role of the molecule in this context.

Minor comments:

GDF15 is a marker of metabolic stress – thus as tumors grow larger, it is expected that levels of GDF15 expression would increase as nutrient availability in the tumor diminishes and the metabolic demands for growth increase. Thus, it would be informative for the authors to provide relative tumor size effects on levels of GDF15 mRNA expression and circulating GDF15 levels.

How does relative bodyweight in tumor bearing animals change over time with GDF15 antibody administration? Only bodyweight at endpoint is provided.

It is unclear what is being blotted for in figure 3C, and the P and M labels are not described. In figure 3F, it seems there is both a loss of parental and mouse GDF15 levels in cells targeted with a guide against GDF15. In the food intake studies in figure 3F, while reaching statistical significance, the differences are extremely marginal and most likely not relevant to the relative weight loss.

In syngeneic experiments, what is the effect of GDF15 neutralization in the context of a KP tumor? This can provide more insight into the effect of the GDF15 antibody in the context of low levels of GDF15.

Version 1:

Reviewer comments:

Reviewer #1

(Remarks to the Author)

In the current revised version of the manuscript, a huge amount of work has been performed to address all my comments/ raised issues in a satisfactorily manner. The manuscript text has been extensively and consistently reorganized, now flowing and reading much better. I see potentially more limitations in the results performed following reviewer 3 comments, however I think it goes over fair peer-review / my assigned tasks.

I only suggest minor changes in order to optimize and standardize the communication of the results.

Grip strength assessments are now included as functional outcome. Please express the values in N (Newton) or mN representing a force, while grams represent static masses. Moreover, Animal force should be normalized by the animal body weight at the beginning of the experiment to account for the variability in animal size. The interindividual body weight variability may also impact on tissue masses, I thus suggest to use the same normalization (initial body weight) for muscle and fat masses. The normalized values may even be presented in a supplementary table, including the initial body weight

along with the tissues analyzed.

Reviewer #2

(Remarks to the Author)

Reviewer #3

(Remarks to the Author)

While the manuscript has been improved by experiments conducted during the revision process, major concerns remain regarding multiple mechanistic aspects of this study at present time. There is a major mechanistic gap as to why elevations in serum GDF-15 do not robustly decrease food intake in this animal model. Additionally, genetic deletion of GDF-15 in tumor cells or blockade through neutralizing antibody treatment in some cases reduces food intake, which runs in direct conflict with our current understanding how GDF-15 promotes weight loss. GDF-15 binds to its sole receptor GFRAL, which is expressed only in the area postrema and nucleus solitary tract of the brainstem, driving weight loss through reductions in food intake and peripheral sympathetic nervous system activity. In line with this understanding, treatment with the anti-GFRAL antibody should induce an increase in food intake through blockage of GDF-15's actions in the CNS, which the authors do not observe. Furthermore, these high-circulating levels of GDF-15 should induce expression of early response genes in the brainstem, such as cFOS, which is a major omission in the manuscript. Multiple groups have identified that GFRAL expression is limited to the brain-stem and is not expressed in peripheral tissues. The lack of rescue of food intake with neutralization of GDF-15/GFRAL signaling axis suggests the rescue effect is not due to direct actions of GDF-15 but through other artifactual pathways.

Additionally, genetic or pharmacological silencing of GDF-15 seems to improve cachexia in every cell line and animal model utilized in this manuscript, regardless of circulating GDF-15 levels, which makes these interventions most likely artifactual. While the authors acknowledge that blockade of GDF-15 may be due to changes in other molecules, understanding how blockade of GDF-15 changes other circulating factors should be included in the manuscript to conclusively support the conclusions of the paper. The findings presented at this point remain too discordant for publication in Nature Communications.

Minor comments:

Treatment of healthy animals with anti-GDF-15 antibodies is not informative, as these animals possess no basal GDF-15 signaling or systemic inflammation. As requested in prior comments previous to resubmission, the authors should examine the effect of GDF-15 in an Lkb1-WT tumor where circulating levels of GDF-15 are marginal and systemic inflammation is present. Rescue in this setting would suggest an artifactual effect of antibody treatment.

Version 2:

Reviewer comments:

Reviewer #3

(Remarks to the Author)

The authors have provided some additional experiments and published work to suggest that perhaps blockade of GDF15 or its receptor GFRAL can prevent wasting without impacting feeding. Although this is an intriguing hypothesis, there is still a major gap on how this is happening and although the authors cite prior studies, they need to address this mechanistic gap.

Given that GFRAL is only expressed in the brainstem, the assumption is that the observed effects of wasting in relation to GDF15 are occurring through the brainstem without food intake being affected. If that is the case the authors should assess neuronal activity by c-FOS staining in the brainstem area postrema (AP) in response to GDF15 or GFRAL blockade. If they do not see any effects in AP activity that suggests that there is GDF15/GFRAL signaling in another tissue/area which they should define. Is there GFRAL expression in the muscle or the fat?

Response to Referees

Dear Referees,

We thank you for your careful analysis of our paper and are very grateful for your constructive comments. We have responded to the comments asked by the reviewers below, and we have incorporated appropriate changes into the manuscript. As a consequence of addressing these comments through additional experimentation, we believe the quality of the manuscript has been strengthened significantly due to the increased generalizability of our findings. Specifically, new **Figure 4** is now the lung orthotopic xenotransplant model which correlates with our multiple subcutaneous models in the manuscript. Furthermore, new **Figure 5** added GDF15 antibody rescue experiments by initiating treatment after 20% fat mass loss. Suppressing cachexia development after it has already initiated increases translational therapeutic implications. Also, in **Figure 8**, we offer additional mechanistic insight showing that *STK11/LKB1*-mutant NSCLC have an increased integrated stress response (ISR) that is critical to the production and processing of GDF15. Lastly, we find similar cachexia biology in both male and female mice. The details of these findings and our direct response to the reviewers' critiques are found below.

Reviewer #1 Comments:

1) “the mechanism linking the **AMPK/LKB1 pathway** modulation with **GDF15 secretion** is missing”: We now have generated additional data that offers significant insight into the role that *STK11/LKB1* plays when made functionally irrelevant in the setting of more global stress sensing mechanisms. Our findings have highlighted that tumors with loss-of-function mutations in *STK11/LKB1* have an exaggerated integrated stress response resulting in the production and processing of stress/cachexia-inducing cytokine GDF15. This activation occurs through an ER stress signaling pathway of PERK, p $\text{eIF}2\alpha$, and ATF4 (see **Figures 8B-C**). Not only does this increase in the integrated stress response cause elevated production of GDF15, it all appears that ATF4 induction also regulates the maturation of GDF15 from a pro to mature/secreted form (see **Figures 8J-M**). This promotion of GDF15 maturation is crucial since it leads to tumor cell specific release of the cytokine, which we have found to subsequently drive wasting. This mechanism becomes even more relevant since there does not appear to be an induction of or contribution from host tissues towards circulating GDF15 concentrations in our *in vivo* tumor cachexia models, prioritizing cell processing of the cytokine in specifically those tumor cells with *STK11/LKB1* alterations. Future efforts will explore the downstream signaling mechanism and molecules critical to all aspects of GDF15 production and processing.

2) *“The introduction is in my opinion too long and resembles partly a result and partly a discussion chapter, failing to provide the state of the art in the studied field in order to understand the following results. Considering that cancer cachexia does not develop in all cancer types, a brief description of the development of cancer cachexia in NSCLC could be added.”*: Introduction has been shortened and made more concise, all changes are now in red font.

3) *“Animal sex is not specified. The authors should clarify which animal sex as chosen and explain the reason behind the choice.”*: As part of new experimentation included in **Supplemental Figure 2**, we verified no differences in cachexia development *in vivo* between male and female mice. *GDF15*-silenced malignancies showed similar tumor weights at experiment end and rescued all cachexia metrics including body weight, fat mass, muscle mass and grip strength in female mice (**Supplemental Figure 2**) similar to those findings in male mice (**Figures 2-3**).

4) *“In order to present homogeneous data for the different models adopted, the tumor and tissue weights at necropsy (at least muscle and adipose tissue) should be included for all the experiments. Considering that the lean mass assessment by ECHO-MRI cannot dissect the contribution of different tissues, and that the enlargement of a tissue could compensate for the reduction of another, the final tissue weights should be presented, even in a supplemental table.”*: Throughout all experiments, we have added additional tumor and cachexia-related metrics. As requested, we have now added tumor weights/areas in **Figures 2C, 4B, 5C, 5J, 6C, 7B** and **Supplementary Figures 2B, 5C, 6C**.

- a) As requested, we have also now added grip strength as a metric of muscle function in cachexia in **Figures 2L** and **4I** and **Supplemental Figure 2I**.
- b) We have also added specific fat pad and muscle weights in **Figures 3A, 3B, 3F, 3G, 4J, 4M, 5N, 5P** and **Supplemental Figures 2J-M**.
- c) From a histological perspective we have also evaluated adipocyte and myocyte cross sectional area in **Figures 3C-E, 3H, 3I, 4K-L, 4N, 6I-J, 6L-N, 7F-G, 7I-J** and **Supplemental Figures 6J-L**.
- d) In addition, we have also added muscle transcriptional changes that associate with cachexia/wasting in **Figures 3J-K**.
- e) To show that cachexia wasting was independent of tumor size, we also added **Figure 1F** comparing tumor sizes vs *GDF15* serum levels in our patient-derived models.

5) *“Figure 1: measuring host contribution to *GDF15* production in immunodeficient mice is in my opinion inappropriate, since the host response to the presence of the cancer is severely impaired.”*: We have now added to our discussion the following statement after we discuss that *GDF15* is secreted from the tumor: “...understanding the limitation of our immunodeficient model is missing part of the immune system.”

Although we agree that immunodeficient models might inhibit the host response to increase GDF15 under stress, most cell types that previously demonstrated GDF15 production (macrophages- PMID: 9326641, granulocytes (PMID: 34381196), and monocytes (PMID: 34381196) are still present in the NOD/SCID model which are deficient in specifically T cell and B cell function.

6) *“The authors describe the association between CC presence and high tumor GDF15 transcript and released peptide, however a recent observational clinical trial suggest the lack of association between cachexia and high GDF15 (PMID: 37115357). Please discuss.”*: The study mentioned by the reviewer employed an antibody directed against full length GDF15, whereas the GDF15 mature protein processed from a pro form is more likely to be released and is known to be the functional end-product identified in circulation. This study also measured GDF15 levels in all comers, not just patients possessing *STK11/LKB1*-mutated tumors. Furthermore, the recently published Tracer-X study on cachexia (PMID: 37045997) showed an association between cachexia development and circulating mature GDF15 in recurrent NSCLC patients with cachexia.

7) *“Figure 2: tumor-free mice untreated, treated with IgG or anti-GDF15 are missing. Since the tumor-bearing mice treated with anti GDF15 gain weight/mass, I think it is important to show whether the anabolic response is present even in healthy controls despite having almost undetectable GDF15 levels.”* Tumor-free mice injected with PBS, IgG. or anti-GDF15 antibody were administered to NOD-SCID mice in new experimentation and showed no cachexia phenotypic differences among cohorts. This is presented in new **Supplemental Figure 3**.

8) *“Figure 3: Daily food intake is not reduced in H1573 tumor bearing mice showing high GDF15 levels and GDF15 blockade has no impact on food intake. The role of GDF15 in cachexia has been mainly described as an anorexigenic molecule, while it is not the case in the current model. I think the authors should better describe the mechanism underlying GDF15 action in the currently adopted model. Surprisingly, when injecting H2122 cells in mice, anorexia is present and silencing GDF15 protects from anorexia, although H2122 cells produce less GDF15 than H1573 (Figure 5). Again, when using H1437 cells (Figure 7 and S5), it looks like that the reduction of GDF15 release obtained by wt LKB1 reconstitution allows to counteract anorexia. Such scattered results are in overt contrast with the authors conclusion stating that ‘Our studies further illustrated that when the GDF15 increases in serum concentrations were not as high, there was less effect on food intake’ (Line 477). Some values of daily food intake seem to be very high, going above 7g/day/mouse (such as in figure 3F). Can the authors explain such high values? Food intake kinetic per group could be an interesting addition. Have the authors checked for food grinding? How many mice were present in each cage? Was any mouse excluded, sacrificed before the established endpoint, producing a change in the feeding behavior of the remaining mice?”*: We do appreciate the earlier version inconsistencies in food intake among different models. But, with more experiments conducted and more animals studied, we did generate fairly consistent food intake findings in the current, modified iteration of the manuscript. Furthermore, we did observe an occasional surge or

dip in food intake that did have to do with certain mice grinding and destroying food that was then no longer measurable and excluded since the food intake findings were no longer interpretable. We also now address food intake changes in the discussion.

9) *“Figure 4: adipocyte and muscle fiber CSA in healthy controls are missing. 3 mice per cohort for histology looks too low and unlikely to provide such statistical differences. I wonder whether the average CSA from each mouse has been used as a single point or each adipocyte/fiber area has been considered as an independent point. The second option is incorrect. In the third panel of Figure 4E, there is an unexpected increase in mice injected with Δ GFP cells; this highlights a possible limitation of having only 3 mice per cohort. The same applies for Figure 5H, where there is an unexpected reduced CSA in Δ GFP.”*: We now have provided cross sectional area and muscle-stained sections for multiple experiments with proper negative controls throughout the paper. From a histological perspective we have also evaluated adipocyte and myocyte cross sectional area in **Figures 3C-E, 3H, 3I, 4K-L, 4N, 6I-J, 6L-N, 7F-G, 7I-J** and **Supplemental Figures 6J-L**.

10) *“Figure 6, panel B: is tumor volume at sacrifice statistically different? Tumor mass, anorexia, serum GDF15 and cachexia look directly correlated.”*: Tumor volume at sacrifice is not statistically different, and neither is tumor mass among KPL groups shown in **Figures 7A** and **B**. We have now added an associative comparison between tumor volume and cachexia factor GDF15 serum levels in **Figure 1F** to show that tumor volume does not associate with GDF15 serum levels. There is no statistical association between serum GDF15 levels and tumor size. Throughout multiple genetic silencing and antibody neutralization experiments we were able to disassociate tumor growth with cachexia phenotype (see **Figures 2, 4, 5, 7, S4, S6**).

11) *“In panel J from the same figure, it looks like that anti GDF15 initially promotes adipose mass accumulation in KPL-bearing mice, followed by a second phase of ‘resistance/ineffectiveness’. A similar phenomenon seems to occur in mice bearing H1792 Δ STK11 tumors (Supplemental figure 4, panel E). Given the potential translational value of the study, the prophylactic nature of the design is not fully convincing, suggesting to perform a therapeutic study to show the impact of anti GDF15 in early cachectic mice (i.e. starting at day 15 or 20 from KPL cell injection). Similarly, a survival (or even a ‘time to humane endpoint’) study would strongly increase the translational value.”*: GDF15 antibody appears to block the cachexia from a preventative perspective (**Figures 5A-G** and **7**) as well as new data showing antibody treatment reversal of the cachexia phenotype once cachexia development has initiated and progressed (see **Figures 5H-P**). During the revision of this manuscript, this same GDF15 neutralization antibody showed significant beneficial effect for cancer patients with high GDF15 levels in a published clinical trial (PMID: 39282907; added to our discussion) and is moving to a phase III randomized trial across multiple primary cancers to determine if a therapeutic indication can be solicited. Our data also shows that disrupting GDF15 in STK11/LKB1 function-deficient NSCLC with low GDF15 levels also prevent cachexia (**Figures 6** and **7**), which expands the utilization of this GDF15 antibody to patients with low circulating GDF15 concentrations if their tumors carry *STK11/LKB1* loss-of-function mutations. We

also verified GDF15 antibody does not non-specifically increase body weight, fat mass, or lean mass by after administration to non-tumor bearing mice (**Supplemental Figure 3**).

12) *“Panel K: first time showing skeletal muscle weight; I would recommend including healthy controls.”*: We added the appropriate healthy controls to **Figures 3A, B, F, G, Figures 4J, M, Figures 5N, P**, and all new experiments.

13) *“Similarly, relative tumor-free body weight and relative tumor-free lean mass for KPL + IgG/GDF15ab are missing.”* Relative tumor free body weight and tumor free lean mass for KPL studies +/- IgG or GDF15 antibody are now presented (**Figure 7**).

14) *“Figure 7: very elegant gene reconstitution experiment, however, the reduced tumor growth rate in reconstituted LKB1 cells does not allow to conclude on the impact on cachexia, given that the same tumor mass developed in a different time frame likely induced a differential adaptation in the hosts, resulting in a different cachectic state.”* *STK11/LKB1* is a known tumor suppressor gene and it is not surprising that there is decreased tumor growth with reconstitution. Although there is a delay in tumor growth, the slope of growth is similar once the tumors actually start growing. This differential growth in **Figure 8** was accounted for by only measuring wasting at equivalent tumor sizes. New data presented in **Supplemental Figure 6** shows the dependence of the H1437 line on tumor-secreted GDF15 (further increasing the generalizability of our findings). Reconstitution of wild-type *STK11/LKB1* into this line decreased the integrated stress response (measured by PERK, p $\text{eIF2}\alpha$, ATF4, and CHOP induction) and decreased production and processing of GDF15 in cell culture which is independent of tumor growth (**Figures 8A-C**).

15) *“Panel D: the x axis of the graph has to be fixed.”* The axis labeling has been corrected in **Figure 8D**.

16) *“I hardly understand the supplemental figure 5, where two experiments almost identical show contrasting or even opposite results. How can 2 weeks of age difference in adult mice produce such different results?”*: As pointed out by the reviewer, food grinding was an issue with the previous data. We have now included experimentation where food grinding is not a complication of the experiment. See new **Figure 8F**.

17) *“The authors state that GDF15 is relevant for muscle contractility. Considering that in this work GDF15 signaling is blocked, an assessment of muscle function (at least grip test and/or treadmill test, in case physiological muscle strength is not a lab commodity) should be included.”* As requested, we have also now added grip strength studies as an additional metric of muscle function in cachexia in **Figures 2L and 4I** and **Supplemental Figure 2I**.

18) *“Minor Comments”*:

- a. *“Is it necessary to have 7 figures in the main text?”*: To address all of the comments from the reviewers, we now actually have 8 figures and added

more supplementary figures. We did remove two figures to lessen the burden on the reader as pointed out by this reviewer. We removed two supplementary figures that were antibody experiments in patient-derived lines since we already have multiple *GDF15* silencing and antibody experiments in multiple models. We believe the remaining figures are essential to the evaluation of *GDF15* in multiple *STK11/LKB1*-mutated models to show the generalizability of our findings.

- b. *“Line 221: despite having ECHO-MRI and CSA data, to state “adipose and muscle atrophy”, the final tissue weights should be reported.”* Based on this suggestion, we have now added fat pad and muscle weights based on a revised experiment for our main H1573 patient-derived cachexia model (**Figures 3A-B, 3F-G**) as well as in the new orthotopic experiments (**Figures 4J and 4M**) and male-female mouse experiment (**Figures S2J-M**).
- c. *“WB: overexposed and low quality of the results.”* It is unclear what figure the reviewer is referencing. We went through and optimized each immunoblot analysis exposure. We intentionally overexposed the *GDF15* and *ATF4* blots in **Figure 8B** to identify changes across a range of glucose levels.
- d. *“Line 87, ‘STK11/LKB1 functional variant’: do the authors refer to an oncogenic mutant?”* We use the word “functional variant” to describe a genetic change that is predicted to alter the function of the expressed protein.
- e. *“Lines 483-485: the authors propose the use of metabolic cages and single-housing to better study food intake alterations associated with changes in GDF15 circulating levels. While single-housing can be an interesting approach to isolate the behavior/contribution of each animal, changing animal housing numerosity can be an independent factor affecting the feeding behavior, possibly determining discrepancies with the observations presented in this study. The authors should mention this limitation in the proposed future approach.: We added a section to the discussion in the metabolic cage section which reads “...with results interrupted understanding singly-housed mice itself might alter food intake.”*
- f. *“Lines 928, 944, 959: please adjust group numerosity reported in the text (doesn’t match with the related graphs).”* Due to the reviewer’s comment that we have too many figures, we removed the figures that were referenced in 944 and 959. We have appropriately corrected the animal numbers in the figure legend as pointed out in line 928.
- g. Multiple clerical/wording changes: Changes made as pointed out by the reviewer.

Reviewer #2 Comments:

- 1) *“I co-reviewed this manuscript with one of the reviewers who provided the listed reports. This is part of the Nature Communications initiative to facilitate training in peer*

review and to provide appropriate recognition for Early Career Researchers who co-review manuscripts.”: Reviewer 2 co-reviewed this manuscript with one of the reviewers who provided the listed reports. See Reviewer 1 or 3 Responses.

Reviewer #3 Comments:

1) *“The study relies predominantly on subcutaneous tumor transplantation of a large number of cells, with tumors often reaching extremely large sizes (<1cm³) before any meaningful weight loss is observed. In human patients, tumor burden is often poorly predictive of weight loss associated with cachexia, with small tumors capable of inducing cachexia. A main concern over the validity of the conclusions of this manuscript is that these subcutaneous tumor models are poorly reflective of human disease, and do not accurately represent the complex interplay between tumor genetics and the tumor microenvironment. In an autochthonous tumor model of LKB1 deficient NSCLC, where tumors develop in the native environment of the lung, levels of GDF15 were shown to be elevated as compared to LKB1 WT tumors but play no part in phenotypes associated with cachexia. The link between LKB1 loss and GDF15, and the role of GDF15 in promoting cachexia in these tumors, should be studied in models of NSCLC where tumors grow in the native environment in the lung, including genetically engineered mouse models and syngeneic transplant studies. Such studies will give insight as to whether the link between GDF15 and LKB1 loss is robust or an artifact of the model chosen.”*: The subcutaneous transplantation model is a well-accepted model to study cachexia. (PMID: 40298389, PMID: 21680814, PMID: 32661391, PMCID: PMC4224962) The subcutaneous model has similar tumorigenesis kinetics as other models, allows investigators to test multiple human-derived models, and enables investigators to distinguish relevant soluble molecule sources of cachexia-relevant factors from human (tumor) vs mouse (host). We have now validated our original findings in multiple subcutaneous xenotransplant models using several patient-derived lines. We also show that subcutaneous injection of the syngeneic KPL model (derived from the TP53/KRAS GEMM mouse model) into immunocompetent mice are also dependent on GDF15 for cachexia development (see **Figures S5 and 7**). To further address the concerns of tumors developing in the native environment of the lung, we collaborated with the Dr. James Kim laboratory (see new added authors) to conduct an orthotopic lung model experiment. This new data is included in **Figure 4**. Despite a similar tumor burden in the lung, the *GDF15*-silenced tumor-bearing mice have decreased serum GDF15 with a corresponding rescue of body weight, adipose tissue, lean mass, and muscle function, to similar levels found in mice that were not administered tumor. The cachexia findings from the orthotopic model correlate with similar findings from the subcutaneous model, we believe strengthening the manuscript significantly.

2) *“There seems to be an intrinsic effect on neutralizing GDF15 even in tumors that do not have high levels of circulating GDF15 (in particular, Supplemental Figure 4). This suggests that treatment with anti-GDF15 antibody is driving the reversal of cachexia phenotypes rather than through blocking the actions of GDF15. To confirm an on target effect of GDF15 neutralization, the anti-GDF15 antibody studies should be complemented*

with treatment of tumors with anti-GFRAL antibody, the only known receptor of GDF15. Additionally, previous studies have indicated that the principal contributor of GDF15 mediated weight loss is from sympathetic output rather than changes in food intake. It would be insightful to see the effect of chemical sympathectomy by 6-OHDA treatment to observe whether this output is playing a role in cachexia in these models.”: [REDACTED]

3) “LKB1 regulates a host of downstream kinases including AMPK and the salt-inducible kinases (SIKs). Previous studies have indicated the tumor suppressive effects of LKB1 are dependent on the SIKs. Additionally, a CRTC2-CREB-SIK axis mediates the inflammatory potential of cells harboring loss of LKB1. It would be informative to see if increase in GDF15 expression is mediated through this same pathway or if other kinases downstream of LKB1 play a role.”: For additional mechanistic insights, please see Reviewer 1 Comment 1. Additionally, we also knocked out CRTC2 or CREB in the H1437 cachexia inducing NSCLC line and determined that the silencing of either gene failed to block GDF15 production in tissue culture (data not included in manuscript). We are currently investigating other downstream signaling molecules including AMPK, SIK, and AMPK-related kinases that may be responsible for transducing STK11/LKB1-LOF cachexia signals, a focus of a subsequent manuscript.

4) “The evidence linking lowering of GDF15 levels to STK11 reconstitution is clear in Figure 7, yet whether this lowering of GDF15 levels is primarily driving the improvement in weight loss is correlative. Studies utilizing GDF15 loss or neutralization should be performed to examine the role of the molecule in this context.”: As requested by the reviewer, we now include data in **Supplemental Figure 6** where we silence GDF15 in the H1437 cell line. As expected, silencing of GDF15 in this same cell line abrogates cachexia development without affecting tumor growth.

5) “Minor Comments”:

- a. “GDF15 is a marker of metabolic stress – thus as tumors grow larger, it is expected that levels of GDF15 expression would increase as nutrient availability in the tumor diminishes and the metabolic demands for growth increase. Thus, it would be informative for the authors to provide relative tumor size effects on levels of GDF15 mRNA expression and circulating GDF15 levels. As requested by the reviewer, we have included new **Figure 1F** which directly compares tumor sizes versus serum GDF15 levels in our patient-derived non-cachexia-inducing and cachexia-inducing lines. There are no statistical associations between tumor size and serum GDF15 levels.
- b. “How does relative bodyweight in tumor bearing animals change over time with GDF15 antibody administration? Only bodyweight at endpoint is provided.”: Due to the limitation of the size of **Figure 5** after adding the rescue experiment, we did not include the body weight change over time. We have included this data for Reviewer 3’s review for both the old prevention experiment as well as the new rescue antibody experiment.

This data is not significantly different than the endpoint Relative Tumor Free Body Weights included in new **Figure 5**.

- c. "It is unclear what is being blotted for in figure 3C, and the P and M labels are not described. It figure 3F, it seems there is both a loss of parental and mouse GDF15 levels in cells targeted with a guide against GDF15.": P indicates proprotein GDF15, M indicates mature GDF15; we clarified these abbreviations in each respective figure legend.
- d. "In the food intake studies in figure 3F, while reaching statistical significance, the differences are extremely marginal and most likely not relevant to the relative weight loss.": We agree with this assessment, and we changed the corresponding result section to "...silencing of GDF15 promoted minimal change in food intake." Furthermore, we expanded the section on food intake regulation in our discussion.
- e. "In syngeneic experiments, what is the effect of GDF15 neutralization in the context of a KP tumor? This can provide more insight into the effect of the GDF15 antibody in the context of low levels of GDF15.": To address the effect of GDF15 antibody on low levels of GDF15, we injected GDF15 antibody into non-tumor bearing mice (new **Figure S3**). There were no significant changes in lean mass, fat mass, or body weight in mice receiving the GDF15 antibody compared to control groups.

Response to Reviewer Referees Letter

We again thank the reviewers for their critiques of our revised manuscript and appreciate their constructive feedback. Although Reviewer 1 felt that we addressed their major concerns, Reviewer 3 has raised an additional concern that was not mentioned in their original critiques regarding the lack of food intake changes in our cancer cachexia models that are dependent on the secretion of GDF15, a potential contradictory finding. We agree with Reviewer 3 that pulsatile injections of recombinant GDF15 into wild-type mice over a short time frame consistently demonstrate food intake decreases by other investigators (PMID: 28953886, 28846099, 28846098, 28846097). However, we disagree that GDF15's role in body weight loss is solely due to a decrease in food intake in pathological states that occur over longer intervals (PMID: 32661391, 37380764, 26672741). For example, injections of recombinant GDF15 into obesity/MASH preclinical models induces body weight loss; however, the major driver of this outcome is through a food intake independent process (PMID: 37380764). More relevant to our current studies, other groups have also shown that tumor cachexia models associated with and even dependent on elevated GDF15 do not show changes in food intake (PMID: 32661391) with manipulation of circulating GDF15 concentrations. This team also directly compared a recombinant GDF15 administration model to a GDF15 dependent tumor cachexia model and observed that the tumor model, unlike the recombinant GDF15 administration model, did not demonstrate any significant changes in food intake (see data below) with alterations in serum GDF15. Clinically relevant, patients with cancer cachexia with elevated serum GDF15 also did not show significant differences in their appetite for food based on a self-assessment survey (PMID: 26672741). Therefore, our studies demonstrating a lack of change in food intake in our cancer cachexia models is consistent with what other groups have observed in pre-clinical models and clinically.

Ultimately, the scope of this manuscript was focused on linking loss-of-function mutations in *STK11/LKB1* in NSCLC with an increase in tumor cell integrated stress response resulting in the tumor secretion of the cachexia factor GDF15 promoting cachexia. The new concern of Reviewer 3 regarding the lack of food intake changes in our tumor cachexia models relative to GDF15 could also be explained by previous work we have conducted studying leptin counter-regulatory host mechanisms. Our group showed previously that the effects of cachexia factors on promoting anorexia, including those observed with IL-6 family members, are counterbalanced by programmed decreases in systemic concentrations of the pro-anorexic molecule leptin as the mice undergo cachexia-induced adipose wasting, culminating in efforts of the host to return food intake back towards unaltered baseline levels (PMID: 30046014). We provide new data in this revised manuscript showing that serum leptin is also significantly reduced in our models concomitant with wasting associated with elevated serum GDF15 concentrations in *STK11/LKB1*-mutated NSCLC bearing mice. The decreased serum leptin provides a possible explanation for why there is no obvious food intake changes in our *STK11/LKB1*-mutant NSCLC cachexia models.

Although understanding exactly how GDF15 triggers body weight loss in a food intake-independent manner in the pathological condition defined by tumor bearing mice is an interesting question, we anticipate the true insight into this mechanism will be clarified in subsequent publications. But, as a means of satisfying the concerns raised by Reviewer 3, we now show in an exploratory manner that muscle sarcolipin (*Sln*)-mediated energy expenditure is potentially critical to GDF15-supported wasting independent of changes in food intake in our preclinical models. A recent paper has highlighted this food intake independent connection between *Sln* and elevated tissue specific energy expenditure in GDF15 models (PMID: 37380764). Another recent publication in a mouse model of mitochondrial myopathy and body weight loss showed that

GDF15 neutralization significantly decreases muscle SIn while rescuing body weight changes (PMID 39976232). A detailed description of these findings, as subfigures, and the appropriate references are detailed below with our direct response to Reviewer 3's additional concerns. We have also modified our discussion accordingly and add this new serum leptin and muscle sarcolipin data to our figures in an effort to satisfy Reviewer 3's remaining concerns.

Reviewer #1 (Remarks to the Author):

“In the current revised version of the manuscript, a huge amount of work has been performed to address all my comments/ raised issues in a satisfactorily manner. The manuscript text has been extensively and consistently reorganized, now flowing and reading much better. I see potentially more limitations in the results performed following reviewer 3 comments, however I think it goes over fair peer-review / my assigned tasks.

I only suggest minor changes in order to optimize and standardize the communication of the results.”

Minor Comments

- 1) *“Grip strength assessments are now included as functional outcome. Please express the values in N (Newton) or mN representing a force, while grams represent static masses. Moreover, Animal force should be normalized by the animal body weight at the beginning of the experiment to account for the variability in animal size. The interindividual body weight variability may also impact on tissue masses, I thus suggest to use the same normalization (initial body weight) for muscle and fat masses. The normalized values may even be presented in a supplementary table, including the initial body weight along with the tissues analyzed.”*

We have converted the values of grip strength in grams to values in N in **Figures 2M, 4I, and S3J**. We also added each mouse's day 0 body weight, measured grip strength (g), transformed grip strength (N), and normalized grip strength (N/g, grip strength/day 0 body weight) in **Supplementary Table 2**. Both transformed grip strength and normalized grip strength data show consistent results that *STK11/LKB1*-mutated NSCLC tumor implantation decrease grip strength compared to non-tumor bearing mice, while tumor *GDF15* silencing rescues the phenotype.

Reviewer #3 (Remarks to the Author):

Major Comments

- 1) *“While the manuscript has been improved by experiments conducted during the revision process, major concerns remain regarding multiple mechanistic aspects of this study at present time. There is a major mechanistic gap as to why elevations in serum GDF-15 do not robustly decrease food intake in this animal model. Additionally, genetic deletion of GDF-15 in tumor cells or blockade through neutralizing antibody treatment in some cases reduces food intake, which runs in direct conflict with our current understanding how GDF-15 promotes weight loss. GDF-15 binds to its sole receptor GFRAL, which is expressed only in the area postrema and nucleus solitary tract of the brainstem, driving weight loss through reductions in food intake and peripheral sympathetic nervous system activity. In line with this understanding, treatment with the anti-GFRAL antibody should induce an increase in food intake through blockage of GDF-15's actions in the CNS,*

which the authors do not observe. Furthermore, these high-circulating levels of GDF-15 should induce expression of early response genes in the brainstem, such as cFOS, which is a major omission in the manuscript. Multiple groups have identified that GFRAL expression is limited to the brain-stem and is not expressed in peripheral tissues. The lack of rescue of food intake with neutralization of GDF-15/GFRAL signaling axis suggests the rescue effect is not due to direct actions of GDF-15 but through other artifactual pathways.”

We are in complete agreement with the reviewer's assertion that the anorexic effects of GDF15 represent a cornerstone of modern cachexia research. Indeed, numerous studies in both preclinical models and human patients have established a strong correlation between elevated circulating GDF15 levels and anorexia (PMID: 28953886, 28846099, 28846098, 28846097, 39282907). However, the regulation of food intake is a complex process, orchestrated by an interplay of central and peripheral signals that govern hunger, satiation, and long-term energy balance. Therefore, we respectfully argue against the over-simplified emphasis of GDF15-GFRAL anorexia as the only factor driving wasting in our cancer cachexia models. This perspective overlooks the complexity of food intake regulation and a vast body of clinical and experimental evidence of non-anorexia mediated GDF15 related body weight loss in pathological states. We agree with Reviewer 3 that pulsatile injections of recombinant GDF15 into wild-type mice over a short time frame consistently demonstrates food intake decreases, as described by other investigators (see **Extended Figure 4A** below from *Nature Medicine* manuscript PMID: 32661391). However, we disagree that GDF15's role in body weight loss is solely due to a decrease in food intake in pathological states that occur over longer/chronic windows of time (PMID: 32661391, 26672741, 37380764).

REDACTED

The landmark study by Suriben et al. in *Nature Medicine* (PMID: 32661391) demonstrated that their tumor cachexia models dependent on elevated GDF15 for regulating wasting also failed to show changes in food intake similar to our data in our current manuscript with GDF regulation genetically and/or pharmacologically. The investigators implanted HT1080 tumor cells into mice and showed that the mice undergo GDF15-dependent cachexia (see extended Figure 3a in their manuscript). There was not a significant difference in food intake in these animals when compared to non-tumor bearing mice (see extended Figure 4B; included above). The authors also directly compared the mouse model of administered recombinant GDF15 (Extended Figure 4A; included above) versus their GDF15-dependent cancer cachexia model (Extended Figure 4B; included above) to show that the cancer cachexia model wasting was independent of net food intake differences. Although the pulsatile injection of GDF15 caused a significant decrease in food intake, the tumor bearing mice that ultimately succumbed to GDF15-dependent cachexia did not have a change in food intake. The authors further developed a monoclonal antibody that acted as a therapeutic antagonistic, 3P10, targeting GDF15's receptor GFRAL. When the authors administrated 3P10 to HT1080 tumor bearing mice, the drug rescued body weight loss (Extended Figure 4C; included above). Similar to our models, there was also a significant rescue in adipose and lean mass tissue loss (Extended Figures 4D and 4F; included above). In this rescue model, however, the investigators did not observe a significant change in food intake in the 3P10

Table 1 Baseline characteristics

Characteristics	Controls	Cancer weight stable	Cancer weight loss	P value
No. of subjects	59	72	58	
Age in years	63.2			0.025
Mean	9.5	66.8 £	63.1	
SD		9.6	7.6	
Cancer type (n (%))				0.2
Lung		20 (28)	21 (36)	
GI		13 (18)	14 (24)	
GU		16 (22)	6 (10)	
Head and neck		1 (1)	3 (5)	
Other		22 (31)	14 (25)	
Stage † (n (%))				0.06
I		12 (17)	2 (3)	
II		10 (14)	13 (22)	
III		19 (26)	18 (31)	
IV		24 (33)	23 (40)	
Chemotherapy regimen (n (%))				0.57
Platinum-based		16 (22)	17 (29)	
Taxanes		7 (10)	6 (10)	
5-Fluorouracil		9 (13)	11 (19)	
Doxorubicin		2 (3)	2 (3)	
Tyrosine kinase inhibitor		4 (6)	3 (5)	
aLBM (kg)				0.01
Mean	18.5*	17.6	16.2	
SD	2.89	3.3	3.2	
FM (kg)				<0.001
Mean	29.9**	25.9	21.6	
SD	7.22	7.7	7.45	
HGS (kg)				0.04
Mean	140	151*	115	
SD	68	45	65	
ASAS appetite (1–10)				0.32
Mean	5.72	5.7	5.23	
SD	1.45	1.72	2.51	

*P < 0.05 compared with C-WL; £ P < 0.05 compared with other groups. **P < 0.001 compared with C-WL; § P < 0.001 compared with controls. †Staging performed only for solid tumours. COPD, chronic obstructive pulmonary disease; GI, gastrointestinal tumours; GU, genito-urinary tumours; aLBM, appendicular lean body mass; FM, fat mass; HGS, handgrip strength; and ASAS, Anderson symptoms assessment scale.

treatment cohort (Extended Figure 4E; included above). These findings of a lack of food intake changes in the treatment of their GDF15-dependent cancer cachexia models are concordant with our genetically-silenced and pharmacologically inhibited cachexia models outlined in this manuscript suggesting that there are other mechanisms in play in these pathologic tumor states that resist or counterbalance the anorexic affects (see below for a plausible mechanism)

Additionally, and more clinically relevant, Garcia and colleagues measured serum inflammatory markers including GDF15 in non-cancer patient controls (59) and cancer patients with (n=58) and without (n=72) weight loss (*Journal of Cachexia, Sarcopenia, and Muscle*, PMID: 26672741). Of the cancer patients, lung cancer was the most common primary tumor (see their Table 1; included on the left). Serum GDF15 was significantly elevated in cancer patients with weight loss compared to those cancer patients without weight loss and non-cancer controls (See Figure 1 in the manuscript). The cancer patients with weight loss also had significant decrements in fat mass, lean mass, and muscle function as judged by grip strength (see their Table 1; included on the left). These patient characteristics are similar to those that we observe in our GDF15-dependent *STK11/LKB1*-mutated patient-derived tumor bearing mice. Importantly, the

cancer patients with weight loss also did not have a significant difference in food intake as judged by appetite scores (see their **Table 1**; included on previous page).

More recently, there is growing evidence in other pathological conditions that the majority of GDF15's weight loss effect is independent of food intake. Injections of recombinant GDF15 into obesity/MASH models induce body weight loss; however, the majority of this weight loss is due to a food intake independent process (*Nature*, PMID: 37380764). To prove this, the investigators pair-fed animals and matched food intake to cohorts of mice that received two doses of GDF15 (see **Figure 1D**; included on the right). The reduced food intake only accounted for half of the body weight changes observed in their MASH model. The investigators were able to show that increased muscle energy expenditure through elevated muscle sarcolipin (*Slc*) accounted for the majority of the food intake independent body weight loss in their GDF15 injection administration model. This finding provides another example of how GDF15 can promote weight loss without ultimately influencing net food intake.

In another set of studies, the Polg^{D257A/D257A} (PLOG) mouse – a mitochondrial myopathy mouse model with decreased body weight, fat mass and fat-free mass phenocopying cachexia characteristics, showed elevated circulating GDF15 levels. Antibody neutralization of this GDF15 rescued mice from this phenotype, concordant with decreases in muscle *Slc* expression (J Cachexia Sarcopenia Muscle, PMID: 39976232). These two findings suggest that GDF15 could induce food-intake independent body weight loss by increasing muscle *Slc* resulting in increased muscle energy expenditure. Neutralization of GDF15 reverses body weight loss and at the same time significantly decreases muscle *Slc*.

Fig. 1 | GDF15 reduces obesity, insulin resistance and NASH independently of reductions in food intake.

d, The percentage change in body mass over time. Data are mean \pm s.e.m. $n = 10$ mice per group, except for GDF15(5 nmol per kg), for which $n = 9$ mice. P values were calculated using two-way ANOVA with Tukey's multiple-comparison test.

B. POLG-GDF15 mAB2 vs POLG-Veh Volcano Plot

FIGURE 5 | GDF15 neutralization alleviated the abnormal gene expression profile in POLG gastrocnemius muscle. (B) Volcano plot of differentially expressed genes in POLG- GDF15 mAB2 versus POLG-Veh gastrocnemius muscle, following transcriptomic analysis. $n = 8$ /group.

To evaluate if muscle *Sln* expression, a surrogate for muscle energy expenditure, associated with body weight loss in our *STK11/LKB1*-mutated cachexia models dependent on tumor secreted GDF15, we analyzed *Sln* mRNA levels by qPCR in both gastronomic and soleus muscles from these animal cohorts. In our isogenic H1573 subcutaneous male cancer cachexia model, *Sln* expression level negatively correlated with body weight, as tumor-bearing mice with parental H1573 or H1573^{ΔGFP} (intact GDF15) had a higher *Sln* expression in gastronomic muscle (see **Figure 3J** in our revised manuscript) and soleus muscle (see **Figure S2** in our revised manuscript) compared to non-tumor (PBS) bearing mice and mice bearing H1573^{ΔGDF15}, which also correlated with increasing muscle loss/atrophy.

Taken together, our data is consistent with what other investigators have observed in pathological states, especially in their cancer cachexia pre-clinical models. Furthermore, the lack of food intake changes is also observed in patients with cancer-associated weight loss with elevated GDF15. In our opinion, our findings of a lack of change in food intake in our GDF15-dependent *STK11/LKB1*-mutated cancer cachexia models are consistent with what others have observed in the field.

2) “Additionally, genetic or pharmacological silencing of GDF-15 seems to improve cachexia in every cell line and animal model utilized in this manuscript, regardless of circulating GDF-15 levels, which makes these interventions most likely artifactual.”

To generalize our findings, we conducted our experiments using several patient-derived and syngeneic NSCLC lines in immunocompromised and immunocompetent mice, respectively. We also completed our studies in both male and female mice. Additionally, we conducted our studies both in subcutaneous and orthotopic models. We not only genetically silenced *GDF15* from multiple patient-derived lines to show that the host cachexia was suppressed, we also targeted GDF15 with neutralizing antibody in these models to further demonstrate the contribution of GDF15 to the cachexia phenotype. Reviewer 3’s original critique stated “to confirm an on-target effect of GDF15 neutralization, the anti-GDF15 antibody studies should be complemented with treatment of tumors with anti-GFRAL antibody, the only known receptor of GDF15.” Therefore, to confirm an on-target effect of GDF15 neutralization, we obtained GFRAL antibody and confirmed that this administration also blocks the GDF15-dependent *STK11/LKB1*-mutated NSCLC host wasting as suggested by Reviewer 3. In our opinion, this comprehensive approach decreases the possibility that these findings are artificial and instead it increases rigor and generalizability of our studies.

3) “While the authors acknowledge that blockade of GDF-15 may be due to changes in other molecules, understanding how blockade of GDF-15 changes other circulating factors should be included in the manuscript to conclusively support the conclusions of the paper. The findings presented at this point remain too discordant for publication in Nature Communications.”

We previously reported that tumor-secreted cachexia factors (such as leukemia inhibitory factor LIF) could cause significant loss of adipose tissue, muscle atrophy, and anorexia (**PMID: 30046014**) over time in preclinical models. At early longitudinal time points of these published tumor cachexia experiments for this PMID, the wild-type C57/BL6 mice injected with cachexia factor LIF demonstrated an early transient state of hypophagia that returned to normophagia as leptin levels decreased as a compensation to the initial adipose loss from the cytokine cachexia signals (**PMID: 30046014**). This reduction in serum leptin concentrations, which translated into a temporary signal promoting net food intake, counterbalanced the cachexia factor anorexia effect. Administration of cachexia factors into *ob/ob* (leptin deficient) and *db/db* (leptin

receptor deficient) mice led to a perpetual change in food intake due to lack of this host compensatory mechanism. These findings and parallel ones observed in the current submitted manuscript ultimately describe why one does not always observe net reduced food intake in tumor cachexia models, likely due to host compensation for the elevated serum GDF15 with a reduction in serum leptin.

To specifically test if this host counterregulatory mechanism is occurring in our *STK11/LKB1*-mutated cachexia models that are dependent on secreted GDF15, we analyzed the serum leptin levels in multiple experiments presented in the manuscript. In our isogenic H1573 subcutaneous male cachexia mouse model, tumor bearing mice with parental H1573 or H1573 Δ GFP had a significant decrease in their serum leptin concentrations compared to non-tumor (PBS) bearing mice (see **Figure 2F** in the revised manuscript). H1573 Δ GDF15 tumor bearing mice, which did not exhibit the cachexia phenotype, had serum concentrations of leptin similar to that of non-tumor (PBS) bearing mice. In our isogenic H1573 subcutaneous female mouse model (see **Figure S3E** in the revised manuscript), we observed a similar trend of decreased serum leptin concentrations in mice injected with H1573 cell lines with GDF15 intact. Serum leptin concentrations were baseline in female H1573 Δ GDF15 bearing mice. We also observed altered serum leptin levels in our orthotopic experiment (see **Figure S8A** in the revised manuscript) and antibody neutralization experiment (see **Figure S8B** in the revised manuscript) with H1573 cell lines commiserate with the level of adipose wasting, and finally in parallel experiments conducted leveraging the H1437 cell line (see **Figure S8C** in the revised manuscript). In all these experiments, animals administered the *STK11/LKB1*-mutated NSCLC Δ GDF15 lines or parental line treated with GDF15 antibody demonstrated a suppressed cachexia phenotype concomitant with a reversal of the reductions in serum leptin concentrations, approaching the control concentrations observed in mice bearing *STK11/LKB1*-mutated NSCLC Δ GFP lines or treated with IgG. This data is consistent with our previous published findings that decreased leptin can counterbalance the decreased food intake changes induced by cachexia factors. This data likely explains the inconsistency of the anorexia phenotype observed in these GDF15-dependent cancer cachexia models.

Minor comments:

- 1) ***“Treatment of healthy animals with anti-GDF-15 antibodies is not informative, as these animals possess no basal GDF-15 signaling or systemic inflammation. As requested in prior comments previous to resubmission, the authors should examine the effect of GDF-15 in an Lkb1-WT tumor where circulating levels of GDF-15 are marginal and systemic inflammation is present. Rescue in this setting would suggest an artifactual effect of antibody treatment.”***

The experiment proposed by the reviewer is subjecting wild-type *STK11/LKB1* tumor bearing mice with GDF15 neutralization antibody to see if there is a “rescue” of the phenotype. We note that neither healthy animals nor *STK11/LKB1*-WT tumor-bearing mice develop cachexia, rendering the hypothetical “rescue” in the suggested experiment irrelevant to our findings. Wild-type *STK11/LKB1* tumor bearing mice do not exhibit a cachexia phenotype to “rescue” since they are similar to non-tumor bearing mice in regard to body weight and composition. However, we appreciate the reviewer’s concern for an “artifactual effect”. Therefore, we have carefully designed our experiments to mitigate this risk of an “artifactual effect” of our findings. Our comprehensive approach included: 1) **Genetic manipulation**; we utilized CRISPR/Cas9 to knock out *GDF15* in several patient-derived NSCLC cell lines (H1573, H1437, H2122, H1944). The resulting isogenic cell lines, lacking GDF15 protein expression, demonstrated a rescue of cancer cachexia in both

male and female mice as well as in subcutaneous and orthotopic models. 2) **Pharmacological inhibition**; we employed a GDF15 antibody to neutralize GDF15 in both syngeneic mouse models and patient-derived NSCLC (H1573) cancer cachexia models through both preventative and rescue experimental paradigms. We also injected these antibodies into non-tumor bearing mice and did not see any changes in body weight and composition. 3) **Genetic rescue and pathway validation**; in an *STK11/LKB1*-mutated NSCLC cell line (H1437), we restored wild-type *STK11/LKB1*. This intervention not only decreased GDF15 production but also prevented cancer cachexia, strongly supporting a key role for the loss-of-function *STK11/LKB1* variant - GDF15 induction pathway as important to NSCLC patients with cachexia. 4) **Specificity confirmation**; as recommended, we performed a GFRAL neutralization antibody experiment using our tumor cachexia mouse models. These findings further confirmed that the observed effects we are seeing are specifically mediated by GDF15 signaling within our *STK11/LKB1*-mutated NSCLC cancer cachexia model. Collectively, these diverse experimental strategies provide strong evidence that our observations are biologically significant and not attributable to antibody artifacts or offsite effects.

Response to Referees

We again thank the reviewers for their critiques of our second revised manuscript and once again appreciate their constructive feedback. We believe this manuscript has greatly improved due to the critiques provided by all reviewers and the additional experimentation completed to address these critiques. Reviewers 1 and 2 had no additional critiques. Reviewer 3 recognizes that our data as well as other investigators' data demonstrate that blockade of GDF15 can prevent wasting independent of food intake changes in cancer cachexia models. However, they also would like this mechanistic gap to be addressed. We did provide additional data to support compensatory mechanisms by the mouse (i.e. changes in serum leptin and muscle sarcolipin levels) to maintain the food intake levels in our previous revision. We also believe that the exact mechanism will require a substantial body of work that is outside of the scope of this current manuscript and therefore will be addressed in a future manuscript. We believe we have completed the scope of this current manuscript which focused on linking loss-of-function mutations in *STK11/LKB1* in NSCLC with an increase in tumor cell integrated stress response resulting in the tumor secretion of the factor GDF15 promoting cachexia.

Reviewer #3 (Remarks to the Author):

Major Comments

“The authors have provided some additional experiments and published work to suggest that perhaps blockade of GDF15 or its receptor GFRAL can prevent wasting without impacting feeding. Although this is an intriguing hypothesis, there is still a major gap on how this is happening and although the authors cite prior studies, they need to address this mechanistic gap. Given that GFRAL is only expressed in the brainstem, the assumption is that the observed effects of wasting in relation to GDF15 are occurring through the brainstem without food intake being affected. If that is the case the authors should assess neuronal activity by c-FOS staining in the brainstem area postrema (AP) in response to GDF15 or GFRAL blockade. If they do not see any effects in AP activity that suggests that there is GDF15/GFRAL signaling in another tissue/area which they should define. Is there GFRAL expression in the muscle or the fat?”

The scope of this current manuscript focuses on linking loss-of-function mutations in *STK11/LKB1* in NSCLC with an increase in tumor cell integrated stress response resulting in the tumor secretion of the factor GDF15 promoting cachexia. We agree there is still a mechanistic gap with targeting GDF15 or its receptor GFRAL to prevent wasting that seems to be independent of food intake changes. We provided both compensatory mechanisms (decrease in serum leptin) as well as GDF15 food intake independent mechanisms (increase of muscle sarcolipin levels) to provide alternative explanations for the lack of observed food intake changes with GDF15 blockade in cancer cachexia models observed by our group and other groups. Although a detailed understanding of this GDF15 mechanism is warranted to uncover its biological relevance independent of cancer cachexia, we believe this will require enough experimentation to support its own manuscript which we plan to proceed with in the future. We had already detailed in the discussion our future “plan to use syngeneic model systems to characterize *STK11/LKB1*-mutant tumor-derived GDF central and peripheral effects leveraging tissue specific knockout models of the GDF15 receptor, GFRAL” which will ultimately address Reviewer 3's comment and suggestion.